# Cross-Modal Knowledge Distillation without Paired Data: Theoretical Foundation and Algorithm

**Trong Khiem Tran** [* 1 2]   **Anh Duc Chu** [* 2]   **Quang Hung Pham** [2]   **Phi Le Nguyen** [2]   **Trong Nghia Hoang** [1]

## Abstract

Cross-modal knowledge distillation (CMKD) studies how a (large) teacher model trained on one type of data (e.g., images) can guide a (smaller) student model building on another type of data (e.g., text/audio). Existing CMKD methods often require paired multi-modal data with aligned semantics, but obtaining such paired data are often costly and impractical. To mitigate this limitation, we develop a new CMKD framework for the more challenging setting where paired data are unavailable. In particular, we establish a cross-modal distributional relationship between teacher and student models which reveals two fundamental quantities governing effective distillation: feature alignment and label alignment. These quantities characterize semantic discrepancy between modalities at the levels of representation and prediction distributions, respectively. Motivated by this insight, we propose a principled framework, with theoretical guarantees, that enables effective cross-modal knowledge distillation by aligning distributions rather than individual samples. Extensive experiments across a wide range of multimodal benchmarks show that our framework is highly effective in both unpaired and paired data settings, improving significantly over prior work.

## 1. Introduction

Knowledge distillation (KD) was first introduced by (Bucilua et al., 2006) and (Hinton et al., 2015) as a mechanism for transferring predictive knowledge from a complex teacher model to a simpler student model. The key motivation is that highly over-parameterized models are often beneficial during training, as the additional capacity provides flexibility for making mistakes and correcting them. The learned knowledge however can often be compressed into a significantly smaller model with minimal loss in predictive performance, making deployment much more efficient. This can be achieved by making the student model mimic the behavior of the teacher model via matching soft predictions.

In classical in-modal settings, both the teacher and student models operate on the same input representation and data modalities. An existing rich literature has advanced in-modal knowledge distillation by improving how knowledge is transferred from teacher to student models. These include contrastive losses for representation alignment (Tian et al., 2020), variance reduction via Bayes-optimal teachers (Menon et al., 2021), gradient-aware adaptive distillation (Zhu & Wang, 2021), decoupled treatment of target and non-target classes (Zhao et al., 2022), multi-level feature review mechanisms (Chen et al., 2021), and relation-based losses capturing inter- and intra-class structure (Huang et al., 2022; Yang et al., 2023). A more comprehensive review of classical in-modal KD methods is provided in Section 5.1.

In contrast, cross-modal knowledge distillation (CMKD) considers more challenging and practical settings where teacher and student models operate on different data modalities, such as transferring knowledge from image-based models to text- or audio-based models. In its distillation process, the teacher and student inputs correspond to different modalities of the same underlying instance, which typically requires paired multimodal data during training. Recent advances have demonstrated the broad applicability of CMKD across diverse domains, including selective bidirectional distillation for bridging modality gaps (C2KD) (Huo et al., 2024), self-supervised representation learning from unlabeled videos (Sarkar & Etemad, 2022), vision–language self-distillation via cross-attention mechanisms (COSMOS) (Kim et al., 2024), and weakly paired transfer between microscopy images and transcriptomics data (XKD) (Bendidi et al., 2025). Despite this progress, existing CMKD methods remain largely dependent on paired or weakly paired multimodal data with sample-level cor-

[1] School of Electrical Engineering and Computer Science, Washington State University, Pullman, US [2] School of Information and Communications Technology, Hanoi University of Science and Technology, Hanoi, Vietnam. Correspondence to: Trong Nghia Hoang <trongnghia.hoang@wsu.edu>, Trong Khiem Tran <khiem.tran@wsu.edu>.

*Proceedings of the 43rd International Conference on Machine Learning*, Seoul, South Korea. PMLR 306, 2026. Copyright 2026 by the author(s).

respondence, which is often costly, difficult to obtain, or impractical in real-world settings where modalities are collected independently or asynchronously. This raises the following fundamental challenge:

**How can we distill knowledge from the teacher model to the student without using sample-level pairing ?**

A natural extension to the unpaired setting is to adapt feature-based knowledge distillation methods by replacing sample-level feature matching with distribution-level feature alignment. However, prior work (Huo et al., 2024) has shown that direct adoption of conventional feature-based distillation methods, such as FitNet (Romero et al., 2014) and ReviewKD (Chen et al., 2021), to cross-modal settings remains ineffective even when paired data are available. This limitation arises because effective cross-modal transfer requires aligning not only representation structures but also predictive semantics across modalities. In the unpaired setting, where semantic sample-level correspondence is entirely absent, this challenge becomes even more severe, as further confirmed by our experimental results (Section 4.2).

To address this challenge, we introduce a principled framework that establishes a provable bound on the generalized student error via two fundamental quantities: (i) **feature alignment** and (ii) **label alignment**. These quantities characterize semantic discrepancy across modalities at the levels of representation and prediction distributions via measuring the cross-modal gap between the teacher model and the student, given a shared representation space. Larger cross-modal discrepancies make effective knowledge transfer substantially more difficult and can lead to suboptimal distillation. This characterization provides actionable guidance for minimizing cross-modal discrepancy, leading to a theoretically grounded framework for algorithm design. Our key contributions are summarized as follows:

**1. Theoretical Analysis**. We develop a theoretical bound that decomposes the distilled student model's generalized error into three components: (i) the teacher's generalized error, which acts as a fixed overhead; (ii) the representation distributional discrepancy between the teacher and student models – **feature alignment**; and (iii) the prediction distributional discrepancy between the teacher and student models – **label alignment**. To the best of our knowledge, this is the first generalization bound for CMKD that reveals a synergistic influence of teacher quality, feature alignment, and label alignment on the student's generalization (Section 2).

**2. Algorithm Design**. Motivated by our theoretical analysis, we develop a practical framework for cross-modal knowledge distillation that focuses on distribution-level alignment which does not require training examples of sample-level pairing. In particular, the developed method enables effective knowledge transfer via minimizing both feature and label alignment. To achieve this, we construct an optimizable surrogate that serves as a medium for selectively distilling teacher knowledge relevant to the student's knowledge. This surrogate is optimized in a bi-level optimization manner by optimizing the student's objective with calibration to feature alignment and label alignment (Section 3).

**3. Empirical Evaluation**. We evaluate our approach on four multimodal benchmarks: AVE (Tian et al., 2018) for event localization; CREMA-D (Cao et al., 2014) and RAVDESS (Livingstone & Russo, 2018) for emotion recognition; and VGGSound (Chen et al., 2020), a large-scale benchmark containing over 200,000 videos spanning more than 300 classes. Across all datasets, our method consistently outperforms recent state-of-the-art (SOTA) baselines in both unpaired and paired CMKD settings, demonstrating its effectiveness and robustness (Section 4).

For interested readers, we also provide a comprehensive literature review of existing KD methods in both in-modal and cross-modal settings in Section 5.

## 2. Theoretical Analysis

We begin by formalizing the problem setting and introducing key notations (Section 2.1). We then establish generalization bounds for the student model under both infinite-data (Section 2.2) and finite-data (Section 2.3) settings.

### 2.1. Problem Setting and Notations

Let $M_T \triangleq (\theta, p_T(y \mid z = \theta(x^T)))$ denote the teacher model which comprises a feature extractor $\theta(x^T)$ and a prediction head $p_T(y \mid z = \theta(x^T))$. The teacher model is pre-trained on a dataset $D^T = (x_i^T, y_i)_{i=1}^{n_T} \sim (\mathcal{X}^T \times \mathcal{Y})$.

Our goal is to distill relevant knowledge from $M_T$ into a student model $M_S \triangleq (\phi, p_S(y \mid \phi(x^S)))$ with feature extractor $\phi(x^S)$ and solution head $p_S$, so that it can generalize better from a student dataset $(x_i^S, y_i)_{i=1}^{n_S} \sim D^S$ with $(x_i^S, y_i) \in \mathcal{X}^S \times \mathcal{Y}$ to unseen data sampled from $(\mathcal{X}^S, \mathcal{Y})$.

We assume the training input to the student model is sampled from new data modalities $\mathcal{X}^S \neq \mathcal{X}^T$ which were not previously seen by the teacher model during its pre-training. To facilitate representation alignment during distillation, we configure the teacher and student feature extractors, $\theta : \mathcal{X}^T \to \mathcal{Z}$ and $\phi : \mathcal{X}^S \to \mathcal{Z}$, to map their respective inputs into a shared embedding space $\mathcal{Z}$.

**Definition 2.1** (Generalized Error). We have the generalized error for the teacher model and the student model under the feature maps $\phi$ and $\theta$ as:

$$\text{err}_T(\theta) \triangleq \mathbb{E}_{D^T(x^T, y)} \left[ -\log p_T \left( y \mid \theta \left( x^T \right) \right) \right], \quad (1)$$

$$\text{err}_S(\phi) \triangleq \mathbb{E}_{D^S(x^S, y)} \left[ -\log p_S \left( y \mid \phi \left( x^S \right) \right) \right], \quad (2)$$

which are the expected (generalized) teacher and student prediction losses over the corresponding data distributions.

**Definition 2.2** (Feature Distribution). The feature distributions $D^S(z)$ and $D^T(z)$ denote the push-forward distributions induced by the student and teacher feature maps, $z = \phi(x^S)$ and $z = \theta(x^T)$, on the marginal input distributions $D^S(x^S)$ and $D^T(x^T)$, respectively.

Let $D^S(y \mid z)$ and $D^T(y \mid z)$ denote the student and teacher feature-label conditional distributions induced by the feature maps $z = \phi(x^S)$ and $z = \theta(x^T)$, respectively.

## 2.2. Asymptotic Performance Bound

We will now provide the generalized performance bound for cross-modal distillation in the asymptotic regime where key quantities (see below) in the bound are computed with respect to the true data distributions. This is equivalent to assuming an infinite amount of data, such that empirical estimates coincide with their population counterparts.

Our main result characterizes the generalized student loss $\text{err}_S(\phi)$ in terms of the following key quantities:

**Overhead.** The generalized teacher loss $\text{err}_T(\theta)$.

**Feature Alignment (FA).** A function of the distributional distance between the student and teacher representation distributions, $D^S(z)$ and $D^T(z)$, induced by $\phi$ and $\theta$.

**Label Alignment (LA).** A function of the distributional distance between the student and teacher predictions, which captures the semantic gap between the student predictor $p_S(y \mid z)$ and the teacher predictor $p_T(y \mid z)$.

An informal statement of our result is stated below.

**Theorem 2.3** (Informal Statement). *Given teacher and student feature maps $\theta$ and $\phi$, we have:*

$$
\begin{aligned}
\text{err}_S(\phi) \;\leq\; & \text{err}_T(\theta) \\
& + \textit{Feature Alignment} + \textit{Label Alignment} \;. \quad (3)
\end{aligned}
$$

This result reveals how the generalization quality of the teacher model and the interplay between **feature alignment** and **label alignment** during distillation will influence the generalized student performance. The teacher generalized performance essentially acts as a fixed overhead. The remaining terms show that feature alignment alone is insufficient for effective cross-modal distillation, as aggressive representation alignment may inadvertently enlarge the semantic discrepancy between the teacher and student prediction distributions. In particular, when alignment induces representations that enlarge this predictive gap, the student model may overfit to misaligned semantic structures, ultimately degrading generalization performance.

This insight is made precise via the below formal definitions and theorem statement (Theorem 2.6).

**Definition 2.4** (Feature Alignment). Let $\Delta$ denote the set of cost metrics $\delta$ on the pre-trained representation space such that the cross-entropy of the teacher prediction,

$$
\ell_\tau(z) \;\triangleq\; -\mathbb{E}_{D^T(y|z)}\Big[\log p_T(y \mid z)\Big], \quad (4)
$$

is $\tau_\delta$-Lipschitz with respect to $\delta$:

$$
\big|\ell_\tau(z_1) - \ell_\tau(z_2)\big| \;\leq\; \tau_\delta \cdot \delta(z_1, z_2) \;. \quad (5)
$$

The feature alignment under the student feature map $\phi$ is

$$
\textbf{FA}(\phi, \theta) \;\triangleq\; \min_{\delta \in \Delta}\Big\{\tau_\delta \cdot W_\delta\big(D^T(z), D^S(z)\big)\Big\} \;. \quad (6)
$$

where $W_\delta$ is Wasserstein-1 distance with cost metric $\delta$.

**Definition 2.5** (Label Alignment). Let $\kappa(y, z)$ denote the label transport kernel between the teacher and student predictors, defined as $\kappa(y, z) \triangleq D^T(y \mid z)/D^S(y \mid z)$.

The label alignment between the teacher predictor $p_T$ and the student predictor $p_S$ is then defined as

$$
\textbf{LA}(p_S, p_T) \;\triangleq\; -\mathbb{E}_{D^S(z,y)}\left[\log\left(\frac{p_S(y \mid z)}{p_T(y \mid z)^{\kappa(y,z)}}\right)\right] \;. \quad (7)
$$

The formal statement of our result can now be stated below.

**Theorem 2.6** (Formal Statement). *Plugging the above definition into the informal statement in Theorem 2.3,*

$$
\text{err}_S(\phi) \;\leq\; \text{err}_T(\theta) \;+\; \textbf{FA}(\phi, \theta) \;+\; \textbf{LA}(p_S, p_T) \;. \quad (8)
$$

*A detailed proof is provided in Appendix A.*

Theorem 2.6 formalizes the earlier intuition by providing a principled characterization of the semantic gap in cross-modal knowledge distillation (CMKD) through both feature alignment (**FA**) and label alignment (**LA**).

## 2.3. Finite-Data Performance Bound

In this section, we analyze the finite-sample regime, where only $n_S$ student samples and $n_T$ teacher samples are available. In particular, we revisit the earlier asymptotic bound by replacing its population-level quantities with their empirical counterparts, leading to the following key quantities characterizing the generalized student loss $\text{err}_S(\phi)$:

**Empirical Label Alignment ($\textbf{LA}_e$).** We define the empirical label alignment ($\textbf{LA}_e$) as a Monte Carlo estimate of the exact label alignment (**LA**) computed from $n_S$ samples:

$$
\textbf{LA}_e(p_S, p_T) \;\triangleq\; -\frac{1}{n_S}\sum_{i=1}^{n_S}\log\left(\frac{p_S(y_i \mid z_i)}{p_T(y_i \mid z_i)^{\kappa(y_i, z_i)}}\right) \;. \quad (9)
$$

**Empirical Feature Alignment ($\textbf{FA}_e$).** We define the empirical feature alignment ($\textbf{FA}_e$) as an empirical estimate of the

population-level feature alignment (**FA**) computed from $n_S$ student samples and $n_T$ teacher samples:

$$\mathbf{FA}_e(\phi, \theta) \triangleq \min_{\delta \in \Delta} \left\{ \tau_\delta W_\delta \left( D_{n_T}^T(\mathbf{z}), D_{n_S}^S(\mathbf{z}) \right) \right\}. \quad (10)$$

The formal statement of our finite-data result is stated below.

**Theorem 2.7** (Formal Statement). *Given the teacher and student feature maps $\theta$ and $\phi$, the following holds with the probability at least $1 - 3\delta$ where $\delta \in (0, 1/3)$:*

$$\mathrm{err}_S(\phi) \leq \mathrm{err}_T(\theta) + \mathbf{FA}_e(\phi, \theta) + \mathbf{LA}_e(p_S, p_T)$$

$$+ \tau_\delta \sqrt{\frac{\log(2/\delta)}{2}} \left( \frac{1}{\sqrt{n_S}} + \frac{1}{\sqrt{n_T}} \right)$$

$$+ O\left( n_S^{-1/s_1} \right) + O\left( n_T^{-1/s_2} \right)$$

$$+ O\left( 2\sqrt{\frac{2d \log(n_S/d)}{n_S}} + \sqrt{\frac{\log(1/\delta)}{2n_S}} \right), \quad (11)$$

*where $s_1$ and $s_2$ are any constants larger than the upper Wasserstein dimensions (Weed & Bach, 2019) of the student and teacher representation distributions, respectively; and $d$ denotes the VC dimension (Mohri et al., 2012) characterizing the complexity of the student hypothesis class. A detailed proof is provided in Appendix B.*

Intuitively, Theorem 2.7 extends the earlier analysis to the practical finite-sample regime, where only a limited number of teacher and student samples are available. Notably, the appearance of the VC dimension reveals a fundamental trade-off between alignment and model complexity: while a more expressive student model may better align with the teacher, increasing its complexity $d$ also enlarges the generalization gap. Consequently, for a fixed number of student samples $n_S$, increasing model capacity does not necessarily improve performance, as the risk of overfitting becomes increasingly dominant, as reflected by the term $O(\sqrt{d/n_S})$.

## 3. Algorithm Design

We now present our proposed Cross-Modal Knowledge Distillation framework, **UCMKD** (**U**niversal **C**ross **M**odal **K**nowledge **D**istillation), designed to address the unpaired-data challenge based on the theoretical insights developed in Section 2. Our framework adopts the bi-level optimization approach in (Finn et al., 2017) that minimizes the generalized student loss (outer optimization) while calibrating the cross-modal semantic gap between teacher and student models (inner optimization). As suggested by our theoretical analysis, this semantic gap is characterized via two complementary quantities: feature alignment (**FA**) and label alignment (**LA**). Accordingly, UCMKD operates through the two-stage workflow illustrated in Figure 1(b). In stage 1, we learn the student encoder $\phi$ by minimizing the representation distributional discrepancy between teacher and student

---

**Algorithm 1** Universal Cross-Modal KD (**UCMKD**)

1: **input:** teacher model $M_T \triangleq (\theta, p_T(y \mid \mathbf{z} = \theta(\boldsymbol{x}^T)))$, student and teacher datasets: $D^S$ and $D^T$, numbers of outer update epochs $n_0$, inner adaptation epochs ($n_1$ and $n_2$), learning rate $\eta$, and hyper-parameters $\lambda_1, \lambda_2$.
2: **output:** student model $M_S \triangleq (\phi, p_S(y \mid \mathbf{z} = \phi(\boldsymbol{x}^S)))$
3: initialize the student encoder and predictor $\phi^0, p_S^0$.
4: **for** $t = 1$ to $n_0$ **do**
5:     $\phi_{\mathrm{tmp}} \leftarrow \phi^{t-1}$ and $p_{\mathrm{tmp}} \leftarrow p_S^{t-1}$
6:     **for** $r = 1$ to $n_1$ **do**
7:        $\phi_{\mathrm{tmp}} \leftarrow \phi_{\mathrm{tmp}} - \eta\lambda_1 \nabla_\phi \ell_{\mathbf{FA}}(\phi_{\mathrm{tmp}})$    # Eq. (14)
8:     **end for**
9:     **for** $r = 1$ to $n_2$ **do**
10:      # see Eq. (15)
11:      $\phi_{\mathrm{tmp}} \leftarrow \phi_{\mathrm{tmp}} - \lambda_2\eta \nabla_\phi \ell_{\mathbf{LA}}(\phi_{\mathrm{tmp}}, p_{\mathrm{tmp}})$
12:      $p_{\mathrm{tmp}} \leftarrow p_{\mathrm{tmp}} - \lambda_2\eta \nabla_p \ell_{\mathbf{LA}}(\phi_{\mathrm{tmp}}, p_{\mathrm{tmp}})$
13:     **end for**
14:     $\phi^t = \phi^{t-1} - \eta\nabla_\phi \mathrm{err}_S(\phi_{\mathrm{tmp}}, p_{\mathrm{tmp}})$    # Eq. (2)
15:     $p_S^t = p_S^{t-1} - \eta\nabla_p \mathrm{err}_S(\phi_{\mathrm{tmp}}, p_{\mathrm{tmp}})$    # Eq. (2)
16: **end for**
17: **return** distilled student model $M_S = (\phi^{n_0}, p_S^{n_0})$

---

models in the latent space (Section 3.1). In stage 2, we optimize both the encoder $\phi$ and prediction head $p_S(y \mid \mathbf{z})$ to minimize the prediction distributional discrepancy between teacher and student models (Section 3.2).

To elaborate on this design, we note that directly optimizing the student objective together with the semantic cross-modal gap from Theorem 2.6, as in conventional CMKD approaches, is often unstable due to the highly coupled interaction between the prediction map $p_S(y \mid \mathbf{z})$ and the feature map $\mathbf{z} = \phi(\boldsymbol{x}^S)$ (see Table 6). In particular, update to the feature extractor $\phi$ simultaneously modifies the alignment between teacher and student distributions while reshaping the student representation landscape $D^S(\mathbf{z})$ on which the predictor $p_S(y \mid \mathbf{z})$ is learned. This moving-target effect substantially complicates the optimization landscape and can lead to unstable convergence. To address this challenge, we adopt a hybrid bi-level optimization approach with a structured two-stage inner-update procedure that calibrates the student loss with respect to the cross-modal semantic gap. In particular, the inner optimization is decomposed into minimizing feature alignment loss (stage 1) and minimizing label alignment loss (stage 2). Figure 1 provides an overview of the proposed framework, which optimizes the student loss via inner adaptation steps used to compute the outer optimization loss in a meta-learning fashion. The pseudo-code of the full algorithm is provided in Algorithm 1.

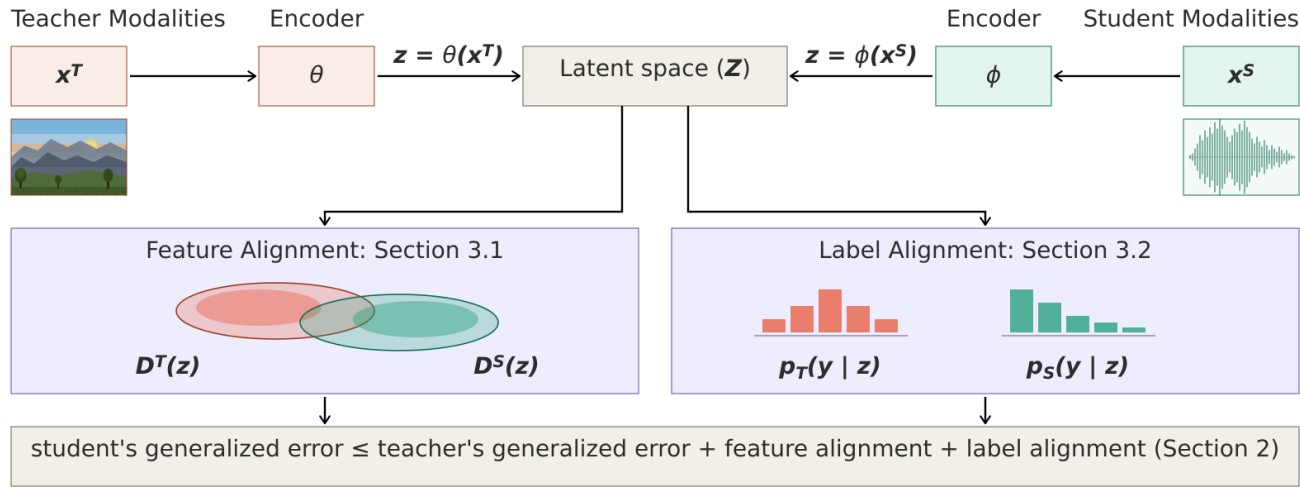

*Figure 1.* **Overview of our UCMKD framework**: The teacher and student encoders map inputs from different modalities into a shared latent space $\mathcal{Z}$. The cross-modal generalization bound decomposes into two distributional quantities: **Feature Alignment**, a Wasserstein distance between the latent distributions $\mathcal{D}^T(\boldsymbol{z})$ and $\mathcal{D}^S(\boldsymbol{z})$ – Section 3.1; and **Label Alignment**, a distance measure between the induced predictive distributions $p_T(y \mid \boldsymbol{z})$ and $p_S(y \mid \boldsymbol{z})$ – Section 3.2. Theorems 2.6 and 2.7 bound the student's generalized error by the sum of teacher error, feature alignment, and label alignment, motivating distribution-level alignment without sample-level pairing.

### 3.1. Feature Alignment Loss

To compute the feature alignment (**FA**), a direct approach is to minimize the optimal transport distance between the empirical student and teacher representation distributions in the shared latent space $\mathcal{Z}$:

$$W_\delta\left(D_{n_S}^S(\boldsymbol{z}), D_{n_T}^T(\boldsymbol{z})\right) = \min_{\pi \in \Pi} \sum_{i=1}^{n_S} \sum_{j=1}^{n_T} \pi_{ij} \delta\left(\boldsymbol{z}_i^S, \boldsymbol{z}_j^T\right) \quad (12)$$

where $\Pi$ denote the set of coupling $\pi$ between $D_{n_S}^S(\boldsymbol{z})$ and $D_{n_T}^T(\boldsymbol{z})$[1] and $\pi_{ij} \triangleq \pi(\boldsymbol{z}_i^S, \boldsymbol{z}_j^T)$. This can be naively achieved via solving a linear program with a computational complexity $O(\max(n_S, n_T)^3)$ (Peyré & Cuturi, 2020).

To reduce this complexity, we instead adopt the following entropic-regularized formulation:

$$\pi^* = \arg\min_{\pi \in \Pi} \left( \sum_{i=1}^{n_S} \sum_{j=1}^{n_T} \pi_{ij} \delta\left(\boldsymbol{z}_i^S, \boldsymbol{z}_j^T\right) + \epsilon H(\pi) \right). \quad (13)$$

This regularized problem can be solved efficiently using the Sinkhorn algorithm with quadratic complexity (Peyré & Cuturi, 2020). In practice, we use the Euclidean transport cost $\delta \triangleq \ell_2$ and the default regularization parameter $\epsilon = 0.1$. The resulting feature alignment loss is defined as

$$\ell_{\textbf{FA}}(\theta, \phi) \quad \triangleq \quad W_{\ell_2}\left(D_{N_S}^S(\boldsymbol{z}), D_{N_T}^T(\boldsymbol{z})\right), \quad (14)$$

where the contribution of feature alignment is controlled by the hyperparameter $\lambda_1$ (Algorithm 1). Here, $D_{n_S}^S(\boldsymbol{z})$ and

---

[1]The corresponding marginals of $\pi(\boldsymbol{z}_i^S, \boldsymbol{z}_j^T)$ over $\boldsymbol{z}_j^T$ and $\boldsymbol{z}_i^S$ correspond to $D_{n_S}^S(\boldsymbol{z}_i^S)$ and $D_{n_T}^T(\boldsymbol{z}_j^T)$.

$D_{n_T}^T(\boldsymbol{z})$ denote the push-forward of the empirical student and teacher input distributions under the corresponding feature maps $\phi$ and $\theta$, respectively.

### 3.2. Label Alignment Loss

The label alignment loss is set to be the empirical $\textbf{LA}_e$ term previously defined in Eq. (9):

$$\ell_{\textbf{LA}}\left(p_S, p_T\right) \triangleq -\frac{1}{n_S} \sum_{i=1}^{n_S} \log\left( \frac{p_S(y_i \mid \boldsymbol{z}_i)}{p_T(y_i \mid \boldsymbol{z}_i)^{\kappa(y_i, \boldsymbol{z}_i)}} \right) \quad (15)$$

where $\kappa(y, \boldsymbol{z}) \triangleq D^T(y \mid \boldsymbol{z})/D^S(y \mid \boldsymbol{z})$.

Intuitively, $\kappa(y, \boldsymbol{z})$ quantifies the agreement between teacher and student prediction distributions, thereby acting as a gating mechanism for **selective knowledge distillation**. A practical estimation procedure for the transport kernel is provided in Appendix D. When the teacher's prediction conflicts with the student's target semantics, particularly when the teacher assigns negligible probability to the target label, the kernel approaches zero, $\kappa(y, \boldsymbol{z}) \simeq 0$. In this regime, the alignment loss naturally reduces to the standard supervised loss of the student model:

$$\lim_{\kappa \to 0} \ell_{\textbf{LA}} = \mathbb{E}_{D^S(\boldsymbol{z}, y)}\left[-\log p_S(y \mid \boldsymbol{z})\right] = \text{err}_S(\phi). \quad (16)$$

This mechanism ensures that, under semantic disagreement, the student model prioritizes its own supervised signal rather than inheriting potentially unreliable teacher guidance. Furthermore, the formulation naturally enables a pseudo-labeling strategy (Nguyen et al., 2020) for minimizing the semantic prediction discrepancy between teacher and student models without requiring paired data, thus improving robustness in unpaired settings.

# 4. Empirical Evaluation

In this section, we present comprehensive experimental results validating the effectiveness of our method under both unpaired-data (Section 4.2) and paired-data (Section 4.3) settings. We further provide ablation studies analyzing parameter sensitivity and performance under data-scarcity scenarios (Section 4.4). Due to limited space, additional experimental results and ablation studies are deferred to Appendix G.

## 4.1. Implementation Details

We evaluate our proposed cross-modal distillation method **UCMKD** on 4 multi-modal datasets which include: (1) **AVE** (Tian et al., 2018) is an audio-visual dataset for audio-visual event localization, which has 28 classes; (2) **CREMA-D** (Cao et al., 2014) is an audio-visual dataset for speech emotion recognition, with 6 categorizations; (3) **RAVDESS** (Livingstone & Russo, 2018) is an audio-visual dataset containing 1,440 emotional utterances with 8 different emotion classes; (4) **VGGsound** (Chen et al., 2020) is a large-scale video dataset containing about 200 000 videos and more than 300 classes covering daily life activities.

**Unpaired Data Simulation.** To simulate the unpaired setting, we apply a stochastic permutation to the original multimodal dataset $\{\mathcal{X}_1, \mathcal{X}_2, \mathcal{Y}\}$. Specifically, we break the instance-level correspondence between modalities by randomly shuffling the indices of one modality relative to the other, resulting in two independent subsets, $\{\mathcal{X}_1, \mathcal{Y}\}$ and $\{\mathcal{X}_2, \mathcal{Y}\}$. This procedure removes sample-level alignment while preserving the marginal distributions of each modality.

**Hyperparameters.** We use the same hyperparameter configuration across all baselines and train each network for 100 epochs with an initial learning rate of $1e-2$. Following (Peng et al., 2022), we adopt ResNet-18 (He et al., 2015) as the backbone architecture for both visual and audio modalities. Additional implementation and hyperparameter details are provided in Appendix F.

## 4.2. Experiment Results on Unpaired Setting

Given the limited literature on knowledge distillation under unpaired settings, we compare our method (**UCMKD**) against: (1) **Cross-Entropy**, which serves as a non-distillation baseline; and (2) **Feature Distillation**, which performs distillation through latent-space representation alignment. Table 1 reports the prediction accuracy of all methods across the evaluated tasks. **UCMKD** consistently achieves the best performance, with an average improvement of approximately $14.3\%$ over **Cross-Entropy** and $7.5\%$ over **Feature Distillation**. Notably, **UCMKD** outperforms **Vanilla KD** (Hinton et al., 2015) on 6 out of 8 tasks, despite **Vanilla KD** benefiting from paired multimodal data, an advantage unavailable to **UCMKD** in our

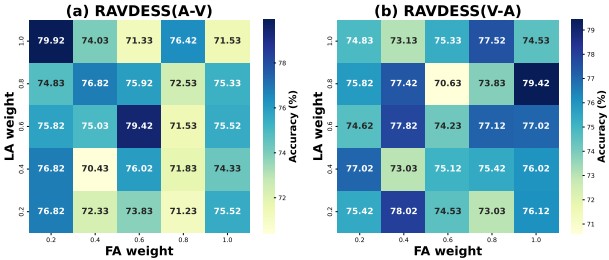

*Figure 2.* Heatmap of the performance of our method (**UCMKD**) on RAVDESS (Livingstone & Russo, 2018) across different values of the hyper-parameters $\lambda_1$ and $\lambda_2$ (Algorithm 1) under (a) audio-to-visual ($A \rightarrow V$) and (b) visual-to-audio ($V \rightarrow A$) settings.

unpaired setting. These results demonstrate the effectiveness of our method in transferring cross-modal knowledge without requiring explicit sample-level correspondence.

## 4.3. Experiment Results on Paired Setting

In this section, we evaluate **UCMKD** under the paired-data knowledge distillation (KD) setting and compare it against **Vanilla KD** (Hinton et al., 2015) and several state-of-the-art baselines, including **C2KD** (Huo et al., 2024), **DKD** (Zhao et al., 2022), **RKD** (Park et al., 2019), **RLD** (Sun et al., 2024), **FitNet** (Romero et al., 2014), and **Review** (Chen et al., 2021). As shown in Table 2, our method consistently outperforms competing baselines on the **AVE**, **RAVDESS**, and **CREMA-D** datasets. In particular, our framework achieves the highest accuracy on 5 out of 8 experimental tasks. This is consistent with the unpaired-setting results in Section 4.2 and further demonstrate the universally robustness of the proposed framework across both paired- and unpaired-data cross-modal distillation settings.

## 4.4. Ablation Studies

**Data Scarcity Scenario.** To evaluate the robustness of our method under data-scarcity conditions, we conduct experiments using reduced training sets. Specifically, we subsample the original training data at ratios of $0.3$, $0.4$, and $0.5$, while retaining the full test set to ensure a consistent evaluation benchmark. Tables 3, 4, and 5 report the performance on the **RAVDESS** dataset of our method and two unpaired-setting baselines: **Cross-Entropy** and **Feature Distillation**. We also include **Vanilla KD** (Hinton et al., 2015) under the paired setting using the same ResNet-18 backbone. The results show that our method consistently achieves the strongest performance across all data-constrained scenarios.

**Parameter Sensitivity.** We conduct a sensitivity analysis to evaluate the effect of the hyperparameters $\lambda_1$ and $\lambda_2$, which control the relative contributions of the Feature Alignment (**FA**) and Label Alignment (**LA**) losses, respectively. Across

*Table 1.* Prediction accuracy on the **AVE**, **RAVDESS**, **CREMA-D**, and **VGGSound** datasets under the unpaired setting, comparing **Cross-Entropy**, **Feature-Based Knowledge Distillation**, and our proposed **UCMKD** framework. For reference, we also report **Vanilla KD** (Hinton et al., 2015) trained under paired supervision. $A \rightarrow V$ and $V \rightarrow A$ denote distillation from audio to visual and visual to audio modalities, respectively. Despite operating without paired supervision, **UCMKD** consistently outperforms unpaired baselines and surpasses the paired **Vanilla KD** baseline on 6 out of 8 experimental tasks.

| Method | AVE | | RAVDESS | | CREMA-D | | VGGsound | |
|---|---|---|---|---|---|---|---|---|
| | $A \rightarrow V$ | $V \rightarrow A$ | $A \rightarrow V$ | $V \rightarrow A$ | $A \rightarrow V$ | $V \rightarrow A$ | $A \rightarrow V$ | $V \rightarrow A$ |
| Teacher | 52.74 | 30.35 | 79.92 | 77.72 | 65.46 | 70.97 | 56.78 | 44.43 |
| Cross Entropy | $27.70 \pm 2.35$ | $50.08 \pm 2.88$ | $65.47 \pm 3.69$ | $70.66 \pm 1.16$ | $71.51 \pm 0.88$ | $61.96 \pm 0.83$ | $41.68 \pm 2.32$ | $54.40 \pm 2.34$ |
| Feature KD | $31.01 \pm 1.04$ | $48.51 \pm 1.49$ | $65.37 \pm 3.20$ | $69.80 \pm 2.96$ | $69.22 \pm 1.38$ | $61.69 \pm 0.69$ | $41.07 \pm 1.84$ | $52.08 \pm 0.83$ |
| Vanilla KD | $29.85 \pm 1.85$ | $49.01 \pm 2.47$ | $67.17 \pm 3.86$ | $73.10 \pm 1.63$ | $\mathbf{72.18 \pm 1.15}$ | $62.45 \pm 0.56$ | $\mathbf{43.35 \pm 0.29}$ | $51.71 \pm 0.69$ |
| **UCMKD** | $\mathbf{34.16 \pm 1.12}$ | $\mathbf{52.24 \pm 1.08}$ | $\mathbf{73.83 \pm 1.25}$ | $\mathbf{74.43 \pm 2.15}$ | $71.64 \pm 0.86$ | $\mathbf{66.67 \pm 1.24}$ | $43.10 \pm 0.38$ | $\mathbf{56.84 \pm 0.47}$ |

*Table 2.* Prediction accuracy on the **AVE**, **RAVDESS**, **CREMA-D**, and **VGGSound** datasets under the paired setting, comparing **Vanilla KD** (Hinton et al., 2015), **C2KD** (Huo et al., 2024), **DKD** (Zhao et al., 2022), **RKD** (Park et al., 2019), **RLD** (Sun et al., 2024), **FitNet** (Romero et al., 2014), **Review** (Chen et al., 2021), and our **UCMKD** framework. $A \rightarrow V$ and $V \rightarrow A$ denote distillation from audio to visual and visual to audio modalities, respectively. **UCMKD** achieves the best performance on 5 out of 8 experimental tasks.

| Method | AVE | | RAVDESS | | CREMA-D | | VGGsound | |
|---|---|---|---|---|---|---|---|---|
| | $A \rightarrow V$ | $V \rightarrow A$ | $A \rightarrow V$ | $V \rightarrow A$ | $A \rightarrow V$ | $V \rightarrow A$ | $A \rightarrow V$ | $V \rightarrow A$ |
| Vanilla KD | $29.85 \pm 1.85$ | $49.01 \pm 2.47$ | $67.17 \pm 3.86$ | $73.10 \pm 1.63$ | $\mathbf{72.18 \pm 1.15}$ | $62.45 \pm 0.56$ | $43.35 \pm 0.29$ | $51.71 \pm 0.69$ |
| RLD | $22.80 \pm 1.22$ | $42.87 \pm 0.82$ | $56.64 \pm 1.98$ | $63.94 \pm 0.85$ | $43.54 \pm 4.59$ | $53.04 \pm 0.67$ | $32.73 \pm 0.66$ | $44.66 \pm 0.70$ |
| RKD | $27.86 \pm 0.70$ | $42.54 \pm 0.54$ | $41.60 \pm 6.33$ | $39.50 \pm 1.53$ | $44.44 \pm 2.79$ | $62.50 \pm 0.87$ | $37.30 \pm 0.49$ | $50.71 \pm 0.40$ |
| DKD | $22.80 \pm 0.65$ | $34.08 \pm 1.66$ | $62.67 \pm 4.27$ | $62.27 \pm 2.57$ | $30.02 \pm 6.11$ | $57.40 \pm 0.38$ | $35.70 \pm 0.28$ | $43.85 \pm 0.29$ |
| C2KD | $33.33 \pm 0.73$ | $47.15 \pm 1.61$ | $56.41 \pm 2.42$ | $\mathbf{82.78 \pm 0.41}$ | $71.50 \pm 0.11$ | $64.43 \pm 0.42$ | $40.90 \pm 0.30$ | $\mathbf{61.90 \pm 0.27}$ |
| FitNet | $25.87 \pm 1.95$ | $49.25 \pm 1.61$ | $68.08 \pm 0.75$ | $69.96 \pm 3.43$ | $70.11 \pm 1.32$ | $65.01 \pm 0.01$ | $37.90 \pm 0.39$ | $57.10 \pm 0.79$ |
| Review | $22.30 \pm 0.62$ | $48.92 \pm 0.65$ | $54.91 \pm 3.20$ | $71.50 \pm 2.00$ | $63.89 \pm 1.68$ | $61.02 \pm 0.54$ | $38.20 \pm 0.47$ | $57.90 \pm 0.79$ |
| **UCMKD** | $\mathbf{33.50 \pm 1.82}$ | $\mathbf{53.07 \pm 0.51}$ | $\mathbf{76.06 \pm 2.28}$ | $75.13 \pm 0.65$ | $70.43 \pm 0.66$ | $\mathbf{66.75 \pm 1.36}$ | $\mathbf{43.70 \pm 0.182}$ | $55.98 \pm 0.38$ |

*Table 3.* Prediction accuracy on the **RAVDESS** dataset using the ResNet-18 backbone with a training subset ratio of 0.5, comparing our method **UCMKD** against the **Cross-Entropy**, **Feature-Based Knowledge Distillation**, and **Vanilla KD** baselines.

| Method | $A \rightarrow V$ | $V \rightarrow A$ |
|---|---|---|
| CE | $59.87 \pm 0.76$ | $71.86 \pm 2.30$ |
| Feature KD | $55.48 \pm 2.85$ | $69.16 \pm 5.35$ |
| Vanilla KD | $57.91 \pm 4.93$ | $69.03 \pm 4.68$ |
| **UCMKD** | $\mathbf{72.76 \pm 5.14}$ | $\mathbf{74.35 \pm 1.27}$ |

*Table 5.* Prediction accuracy on the **RAVDESS** dataset using the ResNet-18 backbone with a training subset ratio of 0.3, comparing our method against the **Cross-Entropy**, **Feature-Based Knowledge Distillation**, and **Vanilla KD** baselines.

| Method | $A \rightarrow V$ | $V \rightarrow A$ |
|---|---|---|
| CE | $54.51 \pm 4.10$ | $68.96 \pm 2.20$ |
| Feature KD | $50.71 \pm 6.76$ | $63.97 \pm 2.20$ |
| Vanilla KD | $50.98 \pm 4.40$ | $68.46 \pm 0.60$ |
| **UCMKD** | $\mathbf{69.03 \pm 2.50}$ | $\mathbf{77.32 \pm 1.34}$ |

*Table 4.* Prediction accuracy on the **RAVDESS** dataset using the ResNet-18 backbone with a training subset ratio of 0.4, comparing our method **UCMKD** against the **Cross-Entropy**, **Feature-Based Knowledge Distillation**, and **Vanilla KD** baselines.

| Method | $A \rightarrow V$ | $V \rightarrow A$ |
|---|---|---|
| CE | $57.14 \pm 3.64$ | $69.93 \pm 3.75$ |
| Feature KD | $50.88 \pm 3.49$ | $71.96 \pm 0.84$ |
| Vanilla KD | $57.68 \pm 1.50$ | $71.63 \pm 0.91$ |
| **UCMKD** | $\mathbf{69.56 \pm 2.17}$ | $\mathbf{78.72 \pm 0.80}$ |

these experiments, the inner-loop optimization steps are fixed to $n_1 = n_2 = 1$ (see Algorithm 1). Figure 2 presents the resulting performance heatmaps (audio-to-visual and visual-to-audio) on the **RAVDESS** dataset (Liu et al., 2022) according to the hyper-parameter grid with different choices for $\lambda_1$ and $\lambda_2$ within 0.2, 0.4, 0.6, 0.8, 1.0. The best performance for audio-to-visual ($A \rightarrow V$) distillation is achieved at $(\lambda_1, \lambda_2) = (0.2, 1.0)$, while visual-to-audio ($V \rightarrow A$)

distillation performs best at $(\lambda_1, \lambda_2) = (1.0, 0.8)$. Notably, even under less favorable hyperparameter configurations, our method consistently outperforms the baselines on $A \rightarrow V$ and remains competitive on $V \rightarrow A$. These results demonstrate the robustness and stability of our **UCMKD** across a broad range of loss-weight configurations.

**Informativeness of Theoretical Bounds.** To empirically validate the informativeness (i.e., tightness) of the theoretical results developed in Section 2, we evaluate the gap between the theoretical bounds and the observed empirical performance across multiple datasets. Figure 3 summarizes the resulting tightness analysis for both the infinite-data setting (Theorem 2.6) and the finite-data regime (Theorem 2.7). Overall, the bounds remain reasonably tight across all datasets, with an average gap of 24.5%. Notably, on the large-scale **VGGSound** dataset containing over 300K+ samples, the gap decreases to 11%, suggesting that the bounds become increasingly informative as data coverage

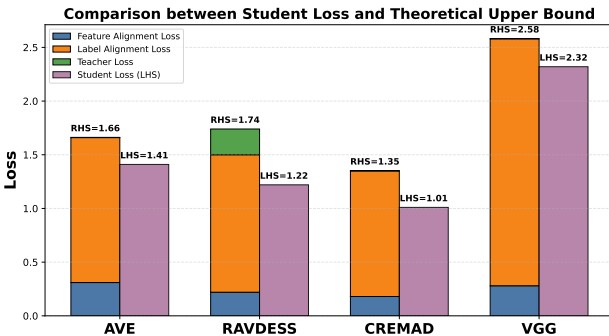

*Figure 3.* Informativeness of the theoretical bound across the **AVE**, **RAVDESS**, **CREMA-D**, and **VGGSound** datasets. The proposed bound remains reasonably tight with an average gap of 24.5%.

*Table 6.* Component-wise ablation results on the **AVE** and **RAVDESS** datasets comparing FA-only (bi-level), LA-only (bi-level), both (w/o bi-level), and our **UCMKD** framework.

| Method | AVE | | RAVDESS | |
|---|---|---|---|---|
| | A→V | V→A | A→V | V→A |
| FA-only (bilevel) | 31.01 | 48.51 | 65.37 | 69.80 |
| LA-only (bilevel) | 30.02 | 48.92 | 67.90 | 69.33 |
| Both (w/o bilevel) | 28.11 | 48.26 | 66.47 | 69.53 |
| **UCMKD** | **34.16** | **52.24** | **73.83** | **74.43** |

grows. This observation is consistent with the theoretical behavior characterized in Theorems 2.6 and 2.7.

**Component-wise Contribution.** Table 6 presents an ablation study on the contributions of feature alignment (FA), label alignment (LA), and the bi-level optimization framework. We observe that FA-only and LA-only variants each achieve competitive performance individually. However, directly combining both alignment losses without the bi-level formulation leads to noticeable performance degradation, highlighting the instability of naive simultaneous optimization (see Section 3). In contrast, **UCMKD**, which integrates both alignment objectives within the proposed bi-level framework, consistently achieves the strongest performance across all settings. These findings align with the theoretical insights of Theorem 2.6 and demonstrate the importance of both alignment components and the bi-level optimization approach.

**Alternative Distribution Distances.** We note that using the $\ell_2$ cost for OT is standard practice in prior works (Damodaran et al., 2018; Tran et al., 2026). Moreover, our theoretical bound is agnostic to the specific choice of cost metric. To further assess this design choice, Table 7 compares the performance of several alternative cost metrics. Overall, all metrics achieve comparable results. Among them, $\ell_2$ achieves marginally better results on 3 out of 4 tasks, making it a reasonable default choice in practice.

**Scalability with Larger Backbones.** To further evaluate

*Table 7.* Comparison of different transport cost metrics, including $\ell_1$, $\ell_2$, and angular distance, on **RAVDESS** and **CREMA-D**.

| Dataset | $\ell_1$ | $\ell_2$ | Angular |
|---|---|---|---|
| RAVDESS (A→V) | **74.23** | 73.83 | 71.03 |
| RAVDESS (V→A) | 73.65 | **74.44** | 73.03 |
| CREMA-D (A→V) | 70.67 | **71.68** | 71.07 |
| CREMA-D (V→A) | 63.52 | **66.59** | 62.41 |

*Table 8.* Scalability evaluation using a ViT-based architecture with ViT-B (patch16-224, 86M parameters) as the teacher and ViT-S (patch16-224, 22M parameters) as the student.

| Method | AVE | | RAVDESS | |
|---|---|---|---|---|
| | A→V | V→A | A→V | V→A |
| Teacher | 75.87 | 70.15 | 90.41 | 89.11 |
| CE | 51.19 | 53.73 | 65.63 | 66.13 |
| Feature KD | 50.96 | 56.22 | 69.83 | 67.73 |
| **UCMKD** | **56.97** | **58.21** | **80.32** | **72.43** |

scalability under more realistic settings, we conduct additional experiments using ViT-based architectures, with ViT-B as the teacher and ViT-S as the student following the protocol in (Addepalli et al., 2024). These models are representative of modern large-scale vision architectures. As shown in Table 8, **UCMKD** consistently achieves the best student performance across all datasets and transfer directions. These results demonstrate that the proposed framework remains effective when transitioning from ResNet-based backbones to substantially larger ViT-based architectures. Combined with the complexity analysis in Appendix E, these findings provide further evidence of the scalability of our approach in practical large-scale settings.

# 5. Related Works

In this section, we provide the literature review of the most relevant works on unimodal KD (Section 5.1) and cross-modal KD (Section 5.2). The detailed formulations of both mentioned settings are provided in Appendix C.

### 5.1. In-Modal Knowledge Distillation

In-Modal Knowledge Distillation (KD) transfers the knowledge of a pretrained teacher model to a student model by minimizing discrepancies between their output predictions or intermediate representations. The seminal work of (Hinton et al., 2015) formulates this as minimizing the Kullback–Leibler (KL) divergence between teacher and student soft predictions, enabling the compression of large models while largely preserving predictive performance. Follow-up work has explored a broad range of alternative mechanisms for effective knowledge transfer. For example, CRD (Tian et al., 2020) introduces contrastive objectives for representation-level distillation, while

SCKD (Zhu & Wang, 2021) adaptively adjusts the distillation process according to gradient similarity between teacher and student models. DKD (Zhao et al., 2022) decomposes KD into target-class and non-target-class distillation to improve flexibility, and Review (Chen et al., 2021) leverages multi-level teacher representations through a feature-review mechanism. DIST (Huang et al., 2022) further develops correlation-based objectives to capture inter-class and intra-class structural relations, while L2D (Yang et al., 2023) extends such relation-based distillation to multi-label classification. SHAKE (Li & JIN, 2022) bridges offline and online KD through auxiliary shadow heads, (Lv et al., 2024) replaces the KL divergence with Wasserstein-distance-based objectives, and RLD (Sun et al., 2024) dynamically refines teacher logits using label information to suppress misleading supervision from incorrect teacher predictions.

Despite their strong empirical performance, these methods predominantly assume that teacher and student models operate on identical training data, thereby overlooking the challenges of Cross-Modal Knowledge Distillation (CMKD), particularly under unpaired settings where explicit sample-level correspondence is unavailable.

### 5.2. Cross-Modal Knowledge Distillation

Cross-Modal Knowledge Distillation (CMKD) extends traditional KD to settings where teacher and student models operate on different data modalities. Early work by Gupta et al. (2015) transfers supervision from a labeled modality to an unlabeled paired modality, while Roheda et al. (2018) employs GANs to transfer knowledge across missing and available modalities. Xue et al. (2021) adapts multimodal networks to unlabeled modalities through pseudo-label sampling from unimodal teachers, and Lee et al. (2023) develops decomposed cross-modal distillation for RGB-based detection using optical-flow supervision.

Other work explores cross-modal transfer for dense indoor prediction (Yun et al., 2023) and theoretical understanding of modality interactions through modality Venn diagrams and modality-focusing hypotheses (Xue et al., 2022). More recent approaches aim to improve cross-modal transfer under increasingly complex settings. C2KD (Huo et al., 2024) introduces selective bidirectional distillation to bridge modality gaps, while (Sarkar & Etemad, 2022) develops a self-supervised framework for cross-modal distillation from unlabeled videos. COSMOS (Kim et al., 2024) further incorporates text-cropping and cross-attention mechanisms for vision-language models, and XKD (Bendidi et al., 2025) explores weakly paired microscopy and transcriptomics data.

Despite these advances, existing CMKD methods still rely heavily on paired or weakly paired multi-modal data, which are often costly or unavailable in realistic settings. Addressing this has been the focus of this paper.

## 6. Limitations and Future Works

While our framework provides a principled formulation and effective algorithm for unpaired CMKD, several directions remain open. First, although our theoretical analysis permits arbitrary transport costs $\delta(\cdot, \cdot)$, we instantiate the framework using the Euclidean metric for simplicity. Learning the transport geometry via adaptive metric learning may further tighten the alignment objective. Second, the proposed bi-level optimization introduces additional computational overhead due to second-order updates or their approximations. Improving optimization efficiency through implicit differentiation or reduced unrolling would further enhance scalability. Third, our results on the large-scale **VGGSound** benchmark suggest that the framework remains effective beyond small curated datasets, motivating future exploration in settings with foundation models where cross-modal transfer must operate without explicit sample-level pairing. Finally, since the proposed framework relies on distribution-level alignment rather than paired supervision, extending it to cross-modal generative modeling under unpaired settings is another promising direction for future follow-up.

## 7. Conclusion

This paper studies cross-modal knowledge distillation under the challenging unpaired setting and proposes **UCMKD**, a principled framework built upon two key components: **Feature Alignment** and **Label Alignment**. We establish both infinite-sample and finite-sample generalization bounds for the student model and develop a practical meta-learning-style optimization framework to realize these objectives. Extensive experiments on **AVE**, **RAVDESS**, **CREMA-D**, and **VGGSound** demonstrate consistent improvements in unpaired settings while remaining competitive with SOTA methods when paired data are available. The proposed formulation also naturally extends beyond prediction task to general distribution-matching losses, opening promising directions for large-scale multi-modal transfer and unsupervised cross-modal generative modeling.

## Acknowledgement

This work utilized GPU compute resources at SDSC and ACES through allocation CIS230391 from the Advanced Cyberinfrastructure Coordination Ecosystem: Services and Support (ACCESS) program (Boerner et al., 2023), which is supported by U.S. National Science Foundation grants #2138259, #2138286, #2138307, #2137603, and #2138296. Trong Nghia Hoang is supported by National Science Foundation CAREER Award IIS-2544071. The authors also acknowledge the compute support from Modal.

## Impact Statement

Our work provides a framework for cross-modal knowledge distillation without requiring paired data, enabling multi-modal transfer in settings where synchronized datasets are unavailable. However, distribution-level alignment might propagate biases or spurious correlations inherited from the teacher model. Careful inspection of the teacher model is important when such systems are deployed in practice.

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

## A. Proof for the infinity data points case.

We have the generalized error for the teacher model and the student model as:

$$\text{err}_T = \mathbb{E}_{D^T(\boldsymbol{x}^T, y)}\Big[-\log p_T(y \mid \boldsymbol{z} = \theta(\boldsymbol{x}^T))\Big] \tag{17}$$

$$\text{err}_S = \mathbb{E}_{D^S(\boldsymbol{x}^S, y)}\Big[-\log p_S(y \mid \boldsymbol{z} = \phi(\boldsymbol{x}^S))\Big] \tag{18}$$

We denote the joint distributions:

$$D^T(\theta(\boldsymbol{x}^T), y) := D^T(\boldsymbol{z}, y) = D^T(\boldsymbol{z})D^T(y \mid \boldsymbol{z}) \tag{19}$$

$$D^S(\phi(\boldsymbol{x}^S), y) := D^S(\boldsymbol{z}, y) = D^S(\boldsymbol{z})D^S(y \mid \boldsymbol{z}) \tag{20}$$

Then, we can rewrite the generalized error of the teacher and the student model as:

$$\text{err}_T = \mathbb{E}_{D^T(\boldsymbol{z}, y)}\Big[-\log(p_T(y \mid \boldsymbol{z}))\Big] = \mathbb{E}_{D^T(\boldsymbol{z})}\mathbb{E}_{D^T(y\mid\boldsymbol{z})}\Big[-\log(p_T(y \mid \boldsymbol{z}))\Big] \tag{21}$$

$$\text{err}_S = \mathbb{E}_{D^S(\boldsymbol{z}, y)}\Big[-\log(p_S(y \mid \boldsymbol{z}))\Big] = \mathbb{E}_{D^S(\boldsymbol{z})}\mathbb{E}_{D^S(y\mid\boldsymbol{z})}\Big[-\log(p_S(y \mid \boldsymbol{z}))\Big] \tag{22}$$

We have:

$$\text{err}_S - \text{err}_T = \mathbf{A} + \mathbf{B} \tag{23}$$

where:

$$\mathbf{A} := \text{err}_S - \mathbb{E}_{D^S(\boldsymbol{z})}\mathbb{E}_{D^T(y\mid\boldsymbol{z})}\Big[-\log(p_T(y \mid \boldsymbol{z}))\Big] \tag{24}$$

$$\mathbf{B} := \mathbb{E}_{D^S(\boldsymbol{z})}\mathbb{E}_{D^T(y\mid\boldsymbol{z})}\Big[-\log(p_T(y \mid \boldsymbol{z}))\Big] - \text{err}_T \tag{25}$$

**1. Bounding A.** We have:

$$\mathbf{A} = \mathbb{E}_{D^S(\boldsymbol{z})}\mathbb{E}_{\mathcal{D}^S(y\mid\boldsymbol{z})}\Big[-\log(p_S(y \mid \boldsymbol{z}))\Big] - \mathbb{E}_{D^S(\boldsymbol{z})}\mathbb{E}_{D^T(y\mid\boldsymbol{z})}\Big[-\log(p_T(y \mid \boldsymbol{z}))\Big] \tag{26}$$

$$= \mathbb{E}_{D^S(\boldsymbol{z})}\Big[\mathbb{E}_{D^S(y\mid\boldsymbol{z})}\Big[-\log(p_S(y \mid \boldsymbol{z}))\Big] - \mathbb{E}_{D^T(y\mid\boldsymbol{z})}\Big[-\log(p_T(y \mid \boldsymbol{z}))\Big]\Big] \tag{27}$$

$$= \mathbb{E}_{D^S(\boldsymbol{z})}\Big[-\sum_{y\in\mathcal{Y}}\Big(D^S(y \mid \boldsymbol{z})\log(p_S(y \mid \boldsymbol{z})) - D^T(y \mid \boldsymbol{z})\log(p_T(y \mid \boldsymbol{z}))\Big)\Big] \tag{28}$$

$$= \mathbb{E}_{D^S(\boldsymbol{z})}\mathbb{E}_{D^S(y\mid\boldsymbol{z})}\Big[-\log(p_S(y \mid \boldsymbol{z})) + \frac{D^T(y \mid \boldsymbol{z})}{D^S(y \mid \boldsymbol{z})}\log(p_T(y \mid \boldsymbol{z}))\Big] \tag{29}$$

With a mild assumption $D^S(y \mid \boldsymbol{z}) > 0$, we denote the label transport kernel $\kappa(y, \boldsymbol{z}) \triangleq \frac{D^T(y\mid\boldsymbol{z})}{D^S(y\mid\boldsymbol{z})}$. Thus, term $\mathbf{A}$ then can be expressed as:

$$\mathbf{A} = \mathbb{E}_{D^S(\boldsymbol{z}, y)}\Big[-\log\Big(\frac{p_S(y \mid \boldsymbol{z})}{p_T(y \mid \boldsymbol{z})^{\kappa(y,\boldsymbol{z})}}\Big)\Big] \tag{30}$$

**2. Bounding B.** We have:

$$\mathbf{B} = \mathbb{E}_{D^S(\boldsymbol{z})}\mathbb{E}_{D^T(y\mid\boldsymbol{z})}\Big[-\log(p_T(y \mid \boldsymbol{z}))\Big] - \mathbb{E}_{D^T(\boldsymbol{z})}\mathbb{E}_{D^T(y\mid\boldsymbol{z})}\Big[-\log(p_T(y \mid \boldsymbol{z}))\Big] \tag{31}$$

$$= \mathbb{E}_{D^S(\boldsymbol{z})}\Big[\ell_\tau(\boldsymbol{z})\Big] - \mathbb{E}_{D^T(\boldsymbol{z})}\Big[\ell_\tau(\boldsymbol{z})\Big] \tag{32}$$

where $\ell_\tau(\boldsymbol{z}) \triangleq \mathbb{E}_{D^T(y\mid\boldsymbol{z})}\Big[-\log(p_T(y \mid \boldsymbol{z}))\Big]$ is the cross-entropy of the teacher prediction as Definition 2.4. For any cost metric $\delta \in \Delta$ such that $|\ell_\tau(\boldsymbol{z}_1) - \ell_\tau(\boldsymbol{z}_2)| \leq \tau_\delta \cdot \delta(\boldsymbol{z}_1, \boldsymbol{z}_2)$, the Kantorovich-Rubinstein duality ascertains that:

$$\mathbf{B} \leq \tau_\delta \mathbf{W}_\delta\Big(D^S(\boldsymbol{z}), D^T(\boldsymbol{z})\Big) \tag{33}$$

where $\mathbf{W}_\delta$ denotes the Wasserstein $-1$ distance with the cost metric $\delta$. Combine Eq. (29) and Eq. (33) we have:

$$\text{err}_S \leq \text{err}_T + \tau_\delta \mathbf{W}_\delta\left(D^S(\boldsymbol{z}), D^T(\boldsymbol{z})\right) + \mathbb{E}_{D^S(\boldsymbol{z},y)}\left[-\log\left(\frac{p_S(y\mid \boldsymbol{z})}{p_T(y\mid \boldsymbol{z})^{\kappa(y,\boldsymbol{z})}}\right)\right] \tag{34}$$

Using Definition 2.4 and Definition 2.5, finally, we complete our proof:

$$\text{err}_S \leq \text{err}_T + \mathbf{FA}(\theta, \phi) + \mathbf{LA}(p_S, p_T) \tag{35}$$

## B. Proof for the finite data points case.

### 1. Rademacher Bounds.

We start with the Rademacher bound (Koltchinskii & Panchenko, 2000), which is stated as follows.

**Rademacher Bounds**. Let $\mathcal{F}$ is the family of functions mapping from $Z$ to $[0,1]$. Then for any $0 < \delta < 1$, with probability at least $1 - \delta$ over sample $S = \{z_1, \cdots, z_n\}$, the following holds for all $f \in \mathcal{F}$:

$$\mathbb{E}[f] \quad \leq \quad \frac{1}{n}\sum_{i=1}^{n} f(z_i) + 2\mathcal{R}_n(\mathcal{F}) + \sqrt{\frac{\log(1/\delta)}{2n}} \tag{36}$$

$$\mathbb{E}[f] \quad \leq \quad \frac{1}{n}\sum_{i=1}^{n} f(z_i) + 2\hat{\mathcal{R}}_S(\mathcal{F}) + 3\sqrt{\frac{\log(2/\delta)}{2n}} \tag{37}$$

Where $\mathcal{R}_n(\mathcal{F})$ and $\hat{\mathcal{R}}_S(\mathcal{F})$ are the Rademacher complexity and the empirical Rademacher complexity.

### 2. Bounding Wasserstein distance.

Feature Alignment (FA) is formulated as **Wasserstein Distance** with the momentum $p = 1$, cost metric $\delta$, and high dimension $d > 1$. For clear notation, we introduce two true probability distributions $\nu$ and $\mu$ with their empirical distributions $\hat{\nu}_n$ and $\hat{\mu}_m$ which provided by $n$ and $m$ data points, respectively. Using the triangle inequality, the Wasserstein distance term can be expressed as:

$$\mathbf{W}(\nu, \mu) \leq \mathbf{W}(\hat{\nu}_n, \hat{\mu}_m) + \mathbf{W}(\nu, \hat{\nu}_n) + \mathbf{W}(\mu, \hat{\mu}_m) \tag{38}$$

Where $\mathbf{W}(\hat{\nu}_n, \hat{\mu}_m)$ can be seen as the empirical estimation of $\mathbf{W}(\nu, \mu)$, which can be directly computed using standard numerical methods. Next, we will explore $\mathbf{W}(\nu, \hat{\nu}_n)$ term as well as $\mathbf{W}(\mu, \hat{\mu}_m)$ term in context **Wasserstein$-1$ Distance**.

For all $n > 0$ and finite momentum $1 \leq p < \infty$, (Weed & Bach, 2019) stated that:

$$\mathbb{P}\left(\mathbf{W}(\nu, \hat{\nu}_n) - \mathbb{E}\left[\mathbf{W}(\nu, \hat{\nu}_n)\right] \geq t\right) \leq \exp(-2nt^2) \tag{39}$$

Thus, with the probability at least $1 - \exp(-2nt^2)$, we have:

$$\mathbf{W}(\nu, \hat{\nu}_n) \leq \mathbb{E}\left[\mathbf{W}(\nu, \hat{\nu}_n)\right] + t \tag{40}$$

We then define $d_p^*(\nu)$ as the upper Wasserstein dimensions (Weed & Bach, 2019). Using Theorem 1 in (Weed & Bach, 2019), given the finite momentum $1 \leq p < \infty$ and $s_1 > d_p^*(\nu)$ is the upper Wasserstein dimension, exist a constant $C_1 > 0$, such that:

$$\mathbb{E}\left[\mathbf{W}(\nu, \hat{\nu}_n)\right] \leq C_1 n^{-1/s_1} \tag{41}$$

Thus, denote $\delta \triangleq \exp(-2nt^2)$ and $0 < \delta < 1$, with the probability at least $1 - \delta$, we have:

$$\mathbf{W}(\nu, \hat{\nu}_n) \leq C_1 n^{-1/s_1} + \sqrt{\frac{\log(2/\delta)}{2n}} \tag{42}$$

Deriving the same steps, with the probability at least $1 - \delta$, we have:

$$\mathbf{W}(\mu, \hat{\mu}_m) \leq C_2 m^{-1/s_2} + \sqrt{\frac{\log(2/\delta)}{2m}} \tag{43}$$

Finally, with the probability at least $1 - 2\delta$, the Wasserstein distance can be bounded by:

$$\mathbf{W}(\nu, \mu) \leq \mathbf{W}(\hat{\nu}_n, \hat{\mu}_m) + C_1 n^{-1/s_1} + C_2 m^{-1/s_2} + \sqrt{\frac{\log(2/\delta)}{2n}} + \sqrt{\frac{\log(2/\delta)}{2m}} \tag{44}$$

where $C_1, C_2, s_1, s_2$ are positive constants, $s_1, s_2$ are larger than the upper Wasserstein dimensions of $\nu$ and $\mu$, respectively.

### 3. Bounding Label Alignment.

Denoting $f(\boldsymbol{z}, y) \triangleq -\log\left(\frac{p_S(y|\boldsymbol{z})}{p_T(y|\boldsymbol{z})^{\kappa(y,\boldsymbol{z})}}\right)$, we then express Label Alignment (**LA**) as:

$$\mathbf{LA} \triangleq \mathbb{E}_{D^S(\boldsymbol{z},y)}\left[-\log\left(\frac{p_S(y \mid \boldsymbol{z})}{p_T(y \mid \boldsymbol{z})^{\kappa(y,\boldsymbol{z})}}\right)\right] = \mathbb{E}_{D^S(\boldsymbol{z},y)}\left[f(\boldsymbol{z}, y)\right] \tag{45}$$

With the mild assumption that the class function $f \in \mathcal{F}$ is upper-bounded by a constraint $C_3 > 0$, we can scale the function $f$ to $[0, 1]$ by dividing by $C_3$ and denote the new class function as $\mathcal{F}/C_3$. Using Rademacher bound (Koltchinskii & Panchenko, 2000), given $0 < \delta < 1$, with the the probability at least $1 - \delta$ over $m$ provided sample, we have:

$$\frac{\mathbb{E}[f]}{C_3} \leq \frac{\hat{E}[f]}{C_3} + 2\mathcal{R}_m(\mathcal{F}/C_3) + \sqrt{\frac{\log(1/\delta)}{2m}} \tag{46}$$

where $\hat{E}[f] \triangleq \frac{1}{m}\sum_{i=1}^m f(\boldsymbol{z}_i, y_i)$ and $\mathcal{R}_m(\mathcal{F}/C_3)$ is Rademacher complexity. By using the property $\alpha \cdot \mathcal{R}(\mathcal{G}) = \mathcal{R}(\alpha \cdot \mathcal{G})$, we have:

$$\mathbb{E}[f] \leq \hat{E}[f] + 2\mathcal{R}_m(\mathcal{F}) + C_3\sqrt{\frac{\log(1/\delta)}{2m}} \tag{47}$$

Let $\Pi_\mathcal{F} : \mathbb{N} \to \mathbb{N}$ be the growth function. Applying Massart's lemma to $\mathcal{R}_m(\mathcal{F})$ (Mohri et al., 2012), we have:

$$\mathcal{R}_m(\mathcal{F}) \leq C_3\sqrt{\frac{2\log \Pi_\mathcal{F}(m)}{m}} \tag{48}$$

Let $d \triangleq \text{VCdim}(\mathcal{F})$ be the VC dimension of the hypothesis class function $\mathcal{F}$. For all $m \in \mathbb{N}$, using Sauer's lemma (Mohri et al., 2012) we have:

$$\Pi_\mathcal{F}(m) \leq \sum_{i=0}^d \binom{m}{i} \tag{49}$$

Then, for all $d \leq n$, we have:

$$\Pi_\mathcal{F}(m) \leq \left(\frac{em}{d}\right)^d \tag{50}$$

Finally, given $0 < \delta < 1$, the function class $f$ is upper-bounded by a constraint $C_3 > 0$, the V-C dimension $d$, with the probability at least $1 - \delta$, we have:

$$\mathbb{E}[f] \leq \hat{E}[f] + 2C_3\sqrt{\frac{2d\log(m/d)}{m}} + C_3\sqrt{\frac{\log(1/\delta)}{2m}} \tag{51}$$

### 4. Bounding the generalized student error on Offline CMKD.

In the Offline CMKD setting, the teacher error is fixed due to the fixed teacher backbone during the distillation process. We can treat the teacher's error as the fixed overhead, then combining E.q (44), and E.q (46), given $0 \leq \delta \leq 1/3$, the teacher and the student empirical distribution $D_{n_T}^T(\boldsymbol{z})$ and $D_{n_S}^S(\boldsymbol{z})$ provided by $n_T$ and $n_S$ data points respectively. Let the Monte Carlo estimation of the Label Alignment (LA) be $\mathbf{LA}_e(p_S, p_T)$, $s_1$ and $s_2$ are larger than the upper-bound Wasserstein dimensions (Weed & Bach, 2019) of the student and teacher representation distribution, respectively, with probability at

least $1 - 3\delta$, we have:

$$\text{err}_S \leq \quad \text{err}_T + \mathbf{LA}_e(p_S, p_T) + 2C_3\sqrt{\frac{2d\log(n_S/d)}{n_S}} + C_3\sqrt{\frac{\log(1/\delta)}{2n_S}} \tag{52}$$

$$+ \quad \tau_\delta\Big(\mathbf{W}\big(D^T_{n_T}(\boldsymbol{z}), D^S_{n_S}(\boldsymbol{z})\big) + C_1 n_S^{-1/s_1} + C_2 n_T^{-1/s_2} + \sqrt{\frac{\log(2/\delta)}{2n_S}} + \sqrt{\frac{\log(2/\delta)}{2n_T}}\Big)$$

We completed our proof.

## C. Knowledge Distillation additional formulation

In this section, we provide additional detailed formulation about Knowledge Distillation in both common settings: unimodal KD (Section C.1) and cross-modal KD (Section C.2).

### C.1. In-Modal Knowledge Distillation

Formally, we consider the $K$-classes classification problem where both the teacher model and the student model receive the same input modality $X$ and produce the logit prediction over $K$ classes. Let $h_\theta(X)$ and $h_\phi(X)$ be the pre-softmax logit of the teacher model and the student model, respectively. Given a temperature $T$, we have the softened predictions as

$$
\begin{aligned}
f_\theta(X;T) &= \text{softmax}(h_\theta(X)/T) \\
f_\phi(X;T) &= \text{softmax}(h_\phi(X)/T)
\end{aligned}
\tag{53}
$$

The student model is trained to minimize the weighted combination of cross entropy loss with respect to the ground-truth labels $Y$ and the distillation loss:

$$\mathcal{L} = \lambda \text{CE}(f_\phi(X), Y) + (1 - \lambda)T^2 \cdot \text{KL}\big(f_\theta(X;T) \,||\, f_\phi(X;T)\big) \tag{54}$$

where KL is the Kullback–Leibler divergence and $\lambda \in [0, 1]$. The objective of the combined loss function is to encourage the small, simple student model to mimic the behavior of large, complex teacher model, thus enabling the compression of the large models while preserving performance (Hinton et al., 2015).

### C.2. Cross-Modal Knowledge Distillation

Cross-modal Knowledge Distillation generalizes the unimodal framework to heterogeneous modalities, allowing a teacher with access to a stronger modality to guide a student with a weaker one. We consider two modalities, denoted by $X_1$ and $X_2$ processed by the teacher and student models, respectively, and a single label $Y$ for both. In this setting, $X_1$ and $X_2$ come from the same instance and have the same label, thus called **paired data** setting. The combined objective function extends from Eq. (54) as (Liu et al., 2022):

$$\mathcal{L} = \lambda \text{CE}(f_\phi(X_2), Y) + (1 - \lambda)T^2 \cdot \text{KL}\big(f_\theta(X_1;T) \,||\, f_\phi(X_2;T)\big) \tag{55}$$

## D. Practical Estimation of the Label Transport Kernel

The theoretical analysis introduces a label-transport kernel $\kappa(y, \boldsymbol{z}) \triangleq D^T(y \mid \boldsymbol{z})/D^S(y \mid \boldsymbol{z})$ to modulate label alignment in the absence of paired samples. In practical implementation, $\kappa$ is only required under the student conditional $y \sim D^S(\cdot \mid \boldsymbol{z})$. For supervised classification, the empirical conditional induced by the labeled student dataset is a Dirac distribution $\hat{D}^S(y \mid \boldsymbol{z}_i) = \delta(y = y_i)$ when given data point $(\boldsymbol{x}_i, y_i)$ under the feature map $\boldsymbol{z}_i = \phi(\boldsymbol{x}_i)$. Consequently, the kernel is evaluated only at the realized label $y_i$, such that

$$\hat{\kappa}_i = \kappa(y_i, \boldsymbol{z}_i) = \frac{\hat{D}^T(y_i \mid \boldsymbol{z}_i)}{\hat{D}^S(y_i \mid \boldsymbol{z}_i)} = \hat{D}^T(y_i \mid \boldsymbol{z}_i) \tag{56}$$

Given a good pre-trained teacher model, we further adopt a plug-in estimator $\hat{D}^T(y_i \mid \boldsymbol{z}_i) = p_T(y_i \mid \boldsymbol{z}_i)$ (pseudo label sampling (Nguyen et al., 2020)). Therefore, $\hat{\kappa}_i = p_T(y_i \mid \boldsymbol{z}_i)$ acts as a teacher–student label-compatibility score: when the teacher assigns low probability to the student's ground-truth label, distillation is downweighted to mitigate negative transfer.

*Table 9.* Per-epoch training time comparison between UCMKD and Feature-based Knowledge Distillation. The relative overhead is computed as the ratio between the training time of UCMKD and Feature Distill.

| Method | AVE | | CREMA-D | | RAVDESS | | VGGSound | |
|---|---|---|---|---|---|---|---|---|
| | A→V | V→A | A→V | V→A | A→V | V→A | A→V | V→A |
| UCMKD | 31.8s | 44.13s | 34.8s | 63.6s | 103.3s | 83.8s | 748.2s | 692.3s |
| Feature KD | 14.7s | 16.3s | 22.5s | 31.9s | 59.4s | 53.0s | 624.8s | 237.9s |
| Relative overhead | 2.16× | 2.76× | 1.54× | 1.99× | 1.73× | 1.58× | 1.20× | 2.91× |

*Table 10.* Scalability evaluation using a ViT-based architecture with ViT-L (ViT-L/16, 300M+ parameters) as the teacher and ViT-S (patch16-224, 22M parameters) as the student.

| Method | AVE | | RAVDESS | |
|---|---|---|---|---|
| | A→V | V→A | A→V | V→A |
| Teacher | 72.64 | 76.87 | 95.50 | 90.57 |
| CE | 51.19 | 53.73 | 65.63 | 66.13 |
| Feature KD | 56.47 | 54.48 | 74.53 | 62.24 |
| **UCMKD** | **59.45** | **61.19** | **88.42** | **77.42** |

# E. Complexity Analysis

As discussed in Section 6, the bilevel optimization in UCMKD introduces additional training cost. However, this overhead mainly comes from a constant-factor increase in the number of student forward/backward passes. It does not introduce a new dependence on the number of training samples beyond the cost already required for computing the alignment objective. Therefore, the scaling behavior with respect to dataset size remains comparable to feature-based knowledge distillation baselines.

**Empirical training cost.** We first report the per-epoch training time of UCMKD and Feature-based KD in Table 9. Across datasets and transfer directions, UCMKD increases the per-epoch training time by a moderate constant factor, ranging from $1.20\times$ to $2.91\times$. This overhead is acceptable given the performance gains over existing baselines reported in Tables 2 and 1.

**Theoretical complexity.** We now analyze the training complexity of UCMKD compared with feature-based KD. Let $N$ denote the number of training samples. We assume the same student and teacher backbones across methods. Let $F_s$ and $F_t$ denote the forward costs of the student and teacher models, respectively, and let $B_s$ denote the backward cost of the student model. Let $W(N)$ denote the cost of computing the Wasserstein distance, e.g., using Sinkhorn iterations with entropic regularization (Peyré & Cuturi, 2020). For feature-based KD, the per-epoch cost can be written as

$$C_{\text{FKD}} = N(F_s + F_t + B_s) + W(N). \tag{57}$$

For UCMKD, we perform one feature-alignment(FA) step and one label-alignment(LA) step per iteration, as used in our experiments. This requires additional student forward/backward passes, while the teacher only needs to be evaluated once. The resulting cost is

$$C_{\text{UCMKD}} = 3N(F_s + B_s) + NF_t + W(N). \tag{58}$$

Therefore,

$$C_{\text{UCMKD}} < 3C_{\text{FKD}}, \tag{59}$$

This shows that UCMKD introduces at most a constant-factor overhead compared with feature-based KD, rather than changing the asymptotic dependence on $N$. Importantly, the optimal transport term $W(N)$ is preserved rather than multiplied by the bilevel procedure. Thus, the main dataset-size-dependent alignment cost remains unchanged. Consequently, UCMKD preserves the scalability of KD-style training while providing substantially improved performance.

*Table 11.* Comparison with recent feature-based KD baselines in the unpaired cross-modal setting.

| Method | AVE | | RAVDESS | | CREMA-D | |
|---|---|---|---|---|---|---|
| | A→V | V→A | A→V | V→A | A→V | V→A |
| NORM | 27.68 | 50.49 | 64.97 | 70.12 | 70.16 | 60.48 |
| REVIEW | 27.12 | 46.27 | 62.61 | 68.42 | 69.76 | 60.62 |
| **UCMKD** | **34.16** | **52.24** | **73.83** | **74.43** | **71.64** | **66.67** |

*Table 12.* Hyperparameter configurations for multimodal datasets **AVE**, **CREMA-D**, **RAVDESS**, **VGGsound**. FA epoch and LA epoch denote the number of epochs minimizing Feature Alignment(**FA**) and Label Alignment(**LA**) in a single distillation epoch. $\lambda_1$ and $\lambda_2$ are the weights of Feature Alignment loss and Label Alignment loss in Algorithm 1.

| Hyperparameter | **AVE** | **CREMA-D** | **RAVDESS** | **VGGsound** |
|---|---|---|---|---|
| Backbone | ResNet-18 | ResNet-18 | ResNet-18 | ResNet-18 |
| Batch size | 64 | 64 | 64 | 64 |
| Epoch | 100 | 100 | 100 | 100 |
| Optimizer | SGD | SGD | SGD | SGD |
| Learning rate | 1e-2 | 1e-2 | 1e-2 | 1e-2 |
| $\lambda_1$ | 1 | 1 | 1 | 1 |
| $\lambda_2$ | 1 | 1 | 1 | 1 |
| FA epoch | 1 | 1 | 1 | 1 |
| LA epoch | 1 | 1 | 1 | 1 |

# F. Implementation detail and Hyperparameters

In this section, we provide more implementation details and hyperparameters to reproduce our empirical results. To ensure fair comparisons, we adopt the same hyperparameters for all compared baselines. The specific hyperparameters are provided in Table 12. All experiments are run on an NVIDIA RTX A6000 GPU, and results are averaged over 5 independent runs. Our official implementation can be found at `https://github.com/Duckduck-05/UCMKD`.

# G. Additional Experimental Results and Ablation Studies

In this section, we present detailed experimental results, including standard deviations, and additional ablation studies of the proposed method that support the empirical analysis in Section 4.

**Scalability with larger backbones.**    Table 13 reports the prediction accuracy on RAVDESS and CREMA-D using ResNet-50 as the backbone, further demonstrating that our method is not tied to a specific architecture and remains effective under a stronger convolutional network. Across the four transfer directions, our method achieves the best performance in three cases, showing consistent robustness across datasets and modality-transfer settings. To further evaluate scalability in more realistic settings, we conduct additional experiments using a ViT-based architecture (ViT-L as the teacher and ViT-S as the student), which is representative of modern large-scale vision models. As shown in Table 10, UCMKD consistently achieves the best student performance across all datasets and transfer directions. These results show that UCMKD remains effective when moving from ResNet-based backbones to substantially larger ViT-based architectures. Together with the complexity analysis (See Appendix E), this provides further evidence that our approach is scalable in realistic settings.

**Compare with modern feature-based KD methods.**    We further evaluate UCMKD against two recent feature-based knowledge distillation methods, REVIEW (Chen et al., 2021) and NORM (Liu et al., 2023). Although these methods were originally designed for paired data knowledge distillation, they can be adapted to the unpaired cross-modal setting and serve as strong feature-based baselines, as discussed in Section 1. As shown in Table 11, UCMKD consistently outperforms both REVIEW and NORM across all datasets and transfer directions. These results further demonstrate that simply applying modern feature-based KD objectives is insufficient for the unpaired cross-modal distillation problem.

**Robustness under distributional mismatch.**    We further evaluate UCMKD under increasingly challenging unpaired settings with distributional mismatch between the teacher and student modalities. Specifically, we consider three levels of

*Table 13.* Prediction accuracy with standard deviation on RAVDESS and CREMA-D with backbone ResNet-50 across different unpaired setting baselines: Cross Entropy, Feature KD, our method, and paired setting Vanilla KD (Hinton et al., 2015). Our method achieves the best performance on 3 out of 4 tasks.

| Method | RAVDESS | | CREMA-D | |
|---|---|---|---|---|
| | $A \rightarrow V$ | $V \rightarrow A$ | $A \rightarrow V$ | $V \rightarrow A$ |
| Teacher | 63.64 | 70.13 | 65.46 | 74.06 |
| CE | $52.25 \pm 0.8$ | $57.58 \pm 2.9$ | $71.33 \pm 1.4$ | $61.16 \pm 0.4$ |
| Feature KD | $50.25 \pm 0.9$ | $72.03 \pm 2.9$ | $69.71 \pm 0.2$ | $62.37 \pm 0.2$ |
| Vanilla KD | $66.67 \pm 2.86$ | $72.60 \pm 1.56$ | $\mathbf{73.20 \pm 1.65}$ | $62.54 \pm 0.17$ |
| **Ours** | $\mathbf{70.83 \pm 4.2}$ | $\mathbf{74.13 \pm 3.5}$ | $72.36 \pm 1.2$ | $\mathbf{66.94 \pm 0.9}$ |

*Table 14.* Performance under increasingly challenging unpaired settings with marginal mismatch, domain shift, and label imbalance on RAVDESS.

| Method | RAVDESS A→V | | | RAVDESS V→A | | |
|---|---|---|---|---|---|---|
| | Easy | Medium | Hard | Easy | Medium | Hard |
| Feature KD | 65.37 | 56.34 | 54.55 | 69.80 | 42.86 | 39.06 |
| CE | 65.47 | 61.04 | 51.45 | 70.66 | 63.24 | 44.96 |
| **UCMKD** | **73.83** | **67.64** | **64.23** | **74.43** | **67.93** | **51.56** |

difficulty: *Easy*, which follows the random permutation setting used in the main text; *Medium*, which introduces marginal mismatch by sampling teacher and student modalities from disjoint sample pools; and *Hard*, which further incorporates modality-specific noise and label imbalance between the teacher and student datasets. As shown in Table 14, both CE and Feature KD suffer substantial performance degradation as the mismatch becomes stronger. In contrast, UCMKD shows a more gradual decline and consistently achieves the best performance across all settings and transfer directions. These results demonstrate that UCMKD is more robust to marginal mismatch, domain shift, and label imbalance in realistic unpaired cross-modal distillation scenarios.

