# OpenReview forum: "Cross-Modal Knowledge Distillation without Paired Data: Theoretical Foundation and Algorithm"
_ICML.cc/2026/Conference — ICML 2026 regular_

### Official Review · Reviewer_CwLu · 2026-03-09

**Soundness:** 3
**Presentation:** 3
**Significance:** 2
**Originality:** 3
**Overall Recommendation:** 4
**Confidence:** 5

**Summary:**

The authors propose a theoretically guaranteed framework that enables effective cross-modal knowledge distillation through distribution alignment instead of individual sample alignment, thereby eliminating the need for data-level pairing. Specifically, this method achieves cross-modal knowledge distillation by minimizing feature and label alignment; it relies on distribution alignment rather than individual sample alignment, thus dispensing with the requirement for data-level pairing.

**Compliance With Llm Reviewing Policy:**

Affirmed.

**Key Questions For Authors:**

No

**Limitations:**

Yes

**Strengths And Weaknesses:**

However, several aspects of this study still require further elaboration and improvement in terms of method design and experimental validation. The specific issues and suggestions are as follows:
1.The Feature Alignment (FA) module aims to achieve precise alignment between the feature representations of the teacher and student models in the shared latent space Z. However, the current description lacks sufficient details on key components such as the optimization objective, loss function design, and iterative update procedure of this module. Its complete optimization mechanism and implementation logic need to be clarified in a clearer and more systematic manner.
2.The core objective of the Label Alignment (LA) module is to synchronize the prediction distributions of the teacher and student models under the same latent embedding z∈Z. Nevertheless, the paper does not explicitly specify the definition, calculation method, and derivation basis of the standard empirical risk for the student model. The acquisition process of this core metric needs to be clearly explained.
3.In the experimental setup, uniform hyperparameter settings are adopted for all baseline models, which lacks sufficient justification. The rationale for such a configuration should be elaborated in detail, including whether it conforms to the optimal settings for each baseline model, so as to avoid compromising the fairness and reliability of the experimental results due to inappropriate hyperparameter configurations.
4.The existing experiments only compare with some traditional methods, resulting in insufficient persuasiveness. It is recommended to supplement quantitative and qualitative comparisons with the mainstream state-of-the-art (SOTA) methods in the field, so as to fully verify the superiority and effectiveness of the proposed approach through more comprehensive experimental results.

---

> ### Author Rebuttal · Authors · 2026-03-31
>
> We sincerely thank the Reviewer for a detailed and thorough review as well as the accept-leaning recommendation. We also appreciate the Reviewer’s constructive feedback and suggestions, which are addressed below.
>
> >The Feature Alignment (FA), the current description lacks sufficient details on key components such as the optimization objective, loss function design, and iterative update procedure of this module.
>
> **Optimization objective.** As defined in Section 3, Feature Alignment (FA) is the function of the distributional distance between student and teacher distribution in the feature space. It has the closed form as the Wasserstein distance with the pre-defined cost metric $\delta$ (see Eq.5)
>
> $FA = \tau_{\delta} W_{\delta}(D^{T}(z), D^{S}(z))$
>
> **Loss function design.** We compute the above Wasserstein distance using the Lagrange method with entropy regularization. It can be computed efficiently using the Sinkhorn algorithm with $\epsilon$  entropic regularization. For practical implementation, we adopt the cost metric as Euclidean distance $\delta = \ell_2$, the hyperparams $\epsilon=0.1$ and $100$ Skinkhorn algorithm iterations. The output of the Sinkhorn algorithm is the optimal transport plan $\pi^*$, such that
>
> $W_{\ell_2}(D^{T}(z), D^{S}(z)) = \sum_{i} \sum_{j} \pi^{*}(i,j) \ell_{2}(z_{i}^{S}, z^{T}_{j})$
>
> **Iterative procedure.** We optimize the FA loss using the standard backpropagation method with respect to the pre-computed optimal transport plan $\pi^*$.
>
> > The Label Alignment (LA), the paper does not explicitly specify the definition, calculation method, and derivation basis of the standard empirical risk for the student model
>
> **Definition.** As defined in Definition 3.5, the Label Alignment (LA) is defined in Eq. (6) which can be computed empirically via Eq. 8:
>
> $LA = - \frac{1}{N} \sum_{i} \log \Big(\frac{p_S(y_i \mid z_i)}{p_{T}(y_i \mid z_i)^{\kappa(y_i, z_i)}} \Big)$
>
> under the feature map $z_i = \phi(x_i)$ with the label transport kernel $\kappa(y_i, z_i) = \frac{D^{T}(y_i \mid z_i)}{D^{S}(y_i \mid z_i)}$, which can be estimated using a pseudo-label sampling strategy [4].
>
> **Calculation method.** Given the data point $(x_i, y_i)$, we first compute the feature map as $z_i = \phi(x_i)$, we then estimate label transport kernel $\kappa(y_i, z_i)$ (see Appendix D), and compute $p_S(y_i \mid z_i), p_T(y_i \mid z_i)$ by feeding $z_i$ to the student and teacher predictor. Given that, we can empirically compute the LA loss.
>
> **Derivation basis of empirical risk.** The generalized risk for student model is defined in Definition 3.1. Replacing the expectation w/ empirical average over the student dataset gives us the corresponding empirical risk.
>
> >In the experimental setup, uniform hyperparameter settings are adopted for all baseline models, which lacks sufficient justification.
>
> We thank the reviewer for the suggestion. Our experiments use a unified training protocol for all methods to ensure a fair comparison: the same backbone, batch size, optimizer family, number of epochs, and learning rate are used across baselines, and the full settings are reported in Section 5.1 and Table 6. Under this common setup, which follows prior practice C2KD [1], our method remains consistently competitive, with results averaged over 5 runs.
>
> >It is recommended to supplement quantitative and qualitative comparisons with the mainstream state-of-the-art (SOTA) methods in the field.
>
> We would like to note that at the submission time in Jan 2026, UCMKD is, to the best of our knowledge, the first work that explores the unpaired data KD setting without requiring the weakly/semantic pairing or expert domain knowledge.
>
> To follow up on the Reviewers' suggestions, we provide additional experiment results with two modern feature-based KD methods: REVIEW [2] and NORM [3], which are not originally designed to work on the unpaired setting; however, they can be repurposed to serve as baselines for this setting. Due to the character limit, we refer the reviewer to the detailed experiment results in our response to reviewer zKfX. The results show that our method also outperforms these new baselines significantly.
>
> [1]  C2kd: Bridging the modality gap for cross-modal knowledge distillation. CVPR 2024
>
> [2]  Distilling Knowledge via Knowledge Review. CVPR, 2021.
>
> [3]  NORM: Knowledge Distillation via N-to-One Representation Matching. ICLR 2023.
>
> [4]  LEEP: A New Measure to Evaluate Transferability of Learned Representations. ICML, 2020.

---

> > ### Author Rebuttal · Reviewer_CwLu · 2026-03-31
> >
> > The author has answered my doubts with great care, and I have no further questions.

---

> > > ### Author Response · Authors · 2026-04-06
> > >
> > > We sincerely thank the Reviewer for acknowledging that the concerns are fully addressed, and for maintaining the positive score.

---

### Official Review · Reviewer_b2pb · 2026-03-10

**Soundness:** 3
**Presentation:** 3
**Significance:** 2
**Originality:** 3
**Overall Recommendation:** 4
**Confidence:** 5

**Summary:**

This paper studies cross-modal knowledge distillation without paired data, a practically important setting that is significantly less explored than standard paired CMKD. The paper proposes a theoretical decomposition of the student generalization error into three components: teacher error, feature alignment, and label alignment. Based on this analysis, the authors introduce UCMKD, a bilevel optimization framework that performs distribution-level distillation instead of sample-level alignment. Concretely, the method aligns teacher and student feature distributions using a Wasserstein-based feature alignment objective and aligns predictive distributions using a label alignment objective with a transport-kernel-based reweighting mechanism.

**Compliance With Llm Reviewing Policy:**

Affirmed.

**Final Justification:**

see above.

**Key Questions For Authors:**

1) How much of the empirical improvement is due to the bilevel optimization scheme itself, as opposed to the two losses alone?

2) Can the authors clarify how well the method handles genuinely unpaired distributions beyond random shuffling of paired datasets?

3) To what extent are the theoretical bounds informative for the implemented method?

4) What is the computational overhead of UCMKD relative to the strongest baselines? Since the method includes both OT computation and bilevel updates, its practical utility depends partly on whether the performance gains justify the added cost.

**Limitations:**

yes

**Strengths And Weaknesses:**

### Strengths

1) The paper tackles an important limitation of existing CMKD methods, namely their reliance on paired or weakly paired multimodal data. This is a highly relevant problem in real-world multimodal learning, where modalities are often collected independently or asynchronously.

2) The decomposition of student error into teacher error + feature alignment + label alignment is intuitive and provides a useful lens for thinking about unpaired CMKD.

3) The method appears consistently strong across several datasets, especially in the unpaired setting. The fact that it can outperform unpaired baselines by a clear margin and, in several cases, even surpass paired Vanilla KD is notable.

4) Although the main contribution is for unpaired data, showing competitiveness in the paired setting strengthens the claim that the proposed framework is broadly useful rather than narrowly specialized.



### Weaknesses
1) Innovation appears somewhat limited at the algorithmic level. While the problem setting is important, the core method is largely a combination of existing ingredients: Wasserstein/optimal-transport-based distribution alignment, KD-style prediction alignment, selective reweighting, and bilevel/meta-learning-style optimization. The paper’s main contribution seems to lie more in the formulation and integration of these components for unpaired CMKD than in introducing fundamentally new technical machinery. As a result, the methodological novelty appears moderate rather than strong.

2) The practical algorithm uses specific simplifications: Euclidean cost for OT, a particular empirical estimator for the label transport kernel, and a specific bilevel optimization scheme. It is not fully clear how essential these choices are, or how tightly the algorithm corresponds to the theory. In particular, some parts of the theory seem to serve more as high-level motivation than as a tight characterization of the implemented method.

3) The quantity $\(\kappa(y,z)=D_T(y|z)/D_S(y|z)\)$ is central to the formulation, but in practice it is approximated in a simplified way. This approximation may be reasonable, but the paper does not fully clarify under what conditions it is reliable, nor whether performance is sensitive to this estimation choice. Since this mechanism is one of the distinctive components of the method, a deeper treatment would strengthen the paper.

4) The unpaired setting is simulated by random permutation of paired datasets. This is a reasonable first step, but it is still a relatively clean version of the unpaired setting. Real unpaired multimodal data may exhibit stronger marginal mismatch, domain shift, or label imbalance across modalities. As a result, the empirical results may somewhat overestimate how robust the method would be in truly unconstrained unpaired scenarios.

5) The paper studies hyperparameter sensitivity, but I would have liked more targeted ablations isolating:
   - feature alignment only vs. label alignment only vs. both,
   - the effect of the transport kernel,
   - the necessity of the bilevel/meta-learning formulation compared with simpler joint optimization,
   - alternative distribution distances.
   These would help verify which parts of the proposed framework are actually responsible for the observed gains.

---

> ### Author Rebuttal · Authors · 2026-03-31
>
> We sincerely appreciate the thoughtful and detailed questions from Reviewer which are addressed below.
>
> **Ablation with alternative distribution distances**
>
> We note that using the $l_2$ cost metric for OT is the standard practice in [1,2] and that the bound holds regardless of the choice of cost metric. We provide below additional experiments with different cost metrics:
>
> |Dataset|$l_1$|$l_2$|angular|
> |-|-|-|-|
> |Ravdess (A→V)|74.226|73.829|71.029|
> |Ravdess (V→A)|73.653|74.428|73.025|
> |CREMA-D (A→V)|70.699|71.671|71.064|
> |CREMA-D (V→A)|63.52|66.582|62.41|
>
> All metrics achieve comparable performance. Among these, $l_2$ achieves marginally better performance on 3/4 tasks, making it a good default choice.
>
> **Transport kernel estimation**
>
> The transport kernel $\kappa( y_i,z_i ) = D^{T}(y_i \mid z_i)/D^{S}(y_i \mid z_i)$ is estimated via approximating $D^S(y_i \mid z_i)$ using empirical samples and $D^T(y_i \mid z_i)$ using the well-established pseudo-label sampling practice in [3] which leverages the teacher model as the plug-in estimator. This is reliable when the teacher model is well-calibrated (often the case with popular pre-trained models).
>
> **Evaluate the effect of transport kernel**
>
> We ablate the effect of the transport kernel via reporting the performance with/without it.
>
> |Method|Cremad||Ravdess||
> |-|-|-|-|-|
> ||A→V|V→A|A→V|V→A||
> |w/oTransport kernel|70.83|61.56|71.30|73.13|
> |UCMKD|71.64|66.67|73.83|74.43|
>
> Without using the transport kernel, the performance decreases consistently across tasks.
>
> **Performance on unpaired distributions with marginal mismatch, domain shift, or label imbalance**
>
> We conduct ablation studies with increasing levels of distributional mismatch between modalities: (i) Easy: random permutation as in the main text; (ii) Medium: marginal mismatch by sampling teacher and student modalities from disjoint sample pools; (iii) Hard: additional perturbations including modality-specific noise and label imbalance between teacher and student datasets.
>
> |Method|Ravdess (A→V) Easy|Ravdess (A→V) Medium|Ravdess (A→V) Hard|Ravdess (V→A) Easy|Ravdess (V→A) Medium|Ravdess (V→A) Hard|
> |-|-|-|-|-|-|-|
> |Feature distill|65.37|56.34|54.55|69.80|42.86|39.06|
> |CE|65.47|61.04|51.45|70.66|63.24|44.96|
> |UCMKD|73.83|67.64|64.23|74.43|67.93|51.56|
>
> The performance of Feature Distill and CE degrades sharply as the setting shifts from Easy to Hard, whereas UCMKD exhibits a much more gradual decline. Across all settings, UCMKD consistently outperforms all baselines by a significant margin. This demonstrates its robustness under more complex unpaired scenarios suggested by the reviewer.
>
> **Ablation FA, LA vs FA+LA & ablation the impact of bilevel optimization**
>
> We ablate the contributions of feature alignment (FA), label alignment (LA), and the bilevel formulation below:
>
> |Method|AVE||Ravdess||
> |-|-|-|-|-|
> ||A→V|V→A|A→V|V→A|
> |FA-only (bilevel)|31.01|48.51|65.37|69.80|
> |LA-only (bilevel)|30.02|48.92|67.90|69.33|
> |Both (w/o-bilevel)|28.11|48.26|66.47|69.53|
> |UCMKD|34.16|52.24|73.83|74.43|
>
> We observe that individually, both FA & LA provide competitive results. However, combining them without the bilevel formulation leads to degraded performance, which underscores the instability of direct joint optimization (see Section 4). In contrast, UCMKD, which integrates both within the bilevel framework, achieves the best performance across all settings. These results align with our result in Thm. 3.6 and confirm the importance of both the alignment terms and bi-level optimization.
>
> **Informativeness of theoretical bounds**
>
> Section 3 provides theoretical guarantees for both the infinite-sample setting (Thm. 3.6) and the finite-sample regime (Thm. 3.7). The informativeness of these bounds increases with more data coverage. To validate this, we evaluate the tightness of the bound across different datasets:
> |Dataset|FA|LA|Teacher Loss|Theoretical Upper Bound (RHS)|Student Loss|Tightness (%)|
> |-|-|-|-|-|-|-|
> |AVE|0.31|1.35|0.002|1.65|1.41|18%|
> |RAVDESS|0.22|1.28|0.240|1.65|1.22|35%|
> |CREMAD|0.18|1.17|0.002|1.35|1.01|34%|
> |VGG|0.28|2.30|0.002|2.58|2.32|11%|
>
> The bound is reasonably tight across all datasets with an average gap of 24.5%. On the large-scale VGG dataset (300K+ samples), it reduces to 11%, indicating that the bound becomes more informative as the data scale grows. This is consistent with our theoretical analysis (Thm. 3.6-3.7).
>
> **Computational overhead of UCMKD**
>
> Due to character limitation, please see our answer to reviewer mgJo’s question, which provides a complexity analysis showing that UCMKD introduces a constant-factor overhead in the number of forward/backward passes, which does not affect its scaling behavior with respect to dataset size.
>
> [1] Optimal Transport for Domain Adaptation. TPAMI, 2017
>
> [2] DeepJDOT: Deep Joint Distribution Optimal Transport for Unsupervised Domain Adaptation. ECCV, 2018
>
> [3] LEEP: A New Measure to Evaluate Transferability of Learned Representations. ICML, 2020

---

> > ### Author Rebuttal · Reviewer_b2pb · 2026-04-03
> >
> > Thanks to the authors' clarification in the rebuttal, I have decided to raise my score to 4.

---

> > > ### Author Response · Authors · 2026-04-06
> > >
> > > We sincerely thank the Reviewer for acknowledging that all concerns are fully addressed and for raising the score to 4.

---

### Official Review · Reviewer_zKfX · 2026-03-11

**Soundness:** 3
**Presentation:** 3
**Significance:** 3
**Originality:** 3
**Overall Recommendation:** 5
**Confidence:** 3

**Summary:**

This paper studies Cross-Modal Knowledge Distillation (CMKD) in the more practically relevant unpaired (no sample-level pairing) scenario. The authors provide a generalization error bound, decomposing the student model's error into: teacher error + Feature Alignment (FA) + Label Alignment (LA). Methodologically, they propose UCMKD: the outer loop minimizes cross-modal representation discrepancies via distribution alignment (using Sinkhorn OT), while the inner loop aligns predictive distributions and performs distillation via a "label transport kernel." Experiments cover datasets such as AVE, CREMA-D, RAVDESS, and VGGSound, claiming state-of-the-art (SOTA) performance in the unpaired setting.

**Compliance With Llm Reviewing Policy:**

Affirmed.

**Final Justification:**

All of my concerns have been addressed during the rebuttal, and I have therefore adjusted my score accordingly.

**Key Questions For Authors:**

Please refer to the weaknesses above.

**Limitations:**

Yes

**Strengths And Weaknesses:**

Strengths
1. Important and realistic problem setting: Relaxing the dependence on paired multimodal data genuinely addresses a key bottleneck in the practical deployment of CMKD.
2. Clear attempt at a theoretical framework: Describing feature distribution discrepancies using Wasserstein/OT and introducing a formal decomposition of LA provides a structured perspective for interpreting the "modality gap."
3. Coverage of multiple audio-visual datasets: The empirical evaluation involves a relatively rich set of targets, demonstrating the authors' effort to validate the method's transferability across datasets.

Weaknesses
1. Severe lack of baselines in the core (unpaired) setting.

2. Please supplement quantitative comparisons with "strong baselines" in the primary unpaired setting (rather than just CE and weak feature distillation).

3. Please provide explicit cost comparisons on VGGSound: wall-clock time, FLOPs, peak memory, and key hyperparameters.

3. What is the essential difference between the LA stage and classic "confidence-weighted distillation / sample reweighting"? Please provide a rigorous mathematical and empirical distinction.

---

> ### Author Rebuttal · Authors · 2026-03-31
>
> We sincerely thank the Reviewer for a detailed and thorough review as well as the accept-leaning recommendation. We also appreciate the Reviewer’s constructive feedback and suggestions, which are addressed below.
>
> >Severe lack of baselines in the unpaired setting. Please supplement quantitative comparisons with "strong baselines" in the primary unpaired setting.
>
> We would like to note that at the submission time, UCMKD is the first work that explores the unpaired data KD setting without requiring the weakly/semantic pairing or expert domain knowledge.
>
> As such, most existing strong CMKD baselines require paired or weakly paired data and are not applicable to the unpaired setting. The comparison in Tab. 1, therefore, can only involve feature distillation baselines that are adapted from KD. This is the most direct and fair comparison in the absence of instance-level correspondence.
>
> To follow up on the Reviewers' suggestions, we further provide additional experiment results with two modern feature-based KD methods: REVIEW [1] and NORM [2], which are not originally designed to work on the unpaired setting; however, they can be repurposed to be the baselines in this problem (as discussed in Section 1). The results show that UCMKD also outperforms both.
>
> |Method|AVE||Ravdess||Cremad||
> |-|-|-|-|-|-|-|
> ||**A→V**|**V→A**|**A→V**|**V→A**|**A→V**|**V→A**|
> |Norm|27.68|50.49|64.97|70.12|70.16|60.48|
> |Review|27.12|46.27|62.61|68.42|69.76|60.62|
> |**UCMKD**|**34.16**|**52.24**|**73.83**|**74.43**|**71.64**|**66.67**|
>
> >Please provide explicit cost comparisons on VGGSound: wall-clock time, FLOPs, peak memory, and key hyperparameters.
>
> As requested, we provide below explicit cost comparisons on VGGSound: wall-clock time each epoch, FLOPs, peak memory. Due to limited resources during rebuttal, we compare against the lightest baseline, Feature Distill; ReviewKD and NORM are more expensive than this baseline.
>
> ||UCMKD||Feature Distill||
> |-|-|-|-|-|
> ||**A→V**|**V→A**|**A→V**|**V→A**|
> |Wall-clock time|748.2s|692.3s|624.8s|237.9s|
> |FLOPs/iter(GFLOPs)|2625.41|3155.87|963.55|963.55|
> |Peak Memory|10.77 GB|16.66 GB|8.11 GB|9.58 GB|
>
> The key hyperparameters are provided in Tab.6, section E. Overall, UCMKD improves accuracy at the cost of higher computation and memory, but the required resources still remain comfortable on standard A6000 GPUs.
>
> >What is the essential difference between the LA stage and classic "confidence-weighted distillation/sample reweighting"?
>
> The key difference lies in both the problem setting and the mathematical role of weighting, as elaborated below.
>
> Confidence-weighted KD methods [3-5] operate in the paired setting and introduce a scalar weight $w_i$ on each sample to modulate the standard pointwise distillation loss:
>
> $L_{WKD}= \sum_i w_i \ell_{\mathrm{KD}}(p_s(\cdot \mid z_i), p_t(\cdot \mid z_i))$
>
> where both student and teacher predictions, $p_s(\cdot\mid z_i)$ and $p_t(\cdot\mid z_i)$, are evaluated on the same input representation $z_i$. The weights $w_i$ are often heuristic functions of confidence or uncertainty. For example, it can be designed to add more weight for the data point with a smaller uncertainty (PAD [3]) or for the hard samples (EA-KD [4]).
>
> In contrast, UCMKD is designed for the unpaired cross-modal setting, where such instance-level correspondence is unavailable. The objective is therefore not sample reweighting, but aligning distributional predictions under a shared feature space. Our formulation is derived from a generalization bound (Theorem 3.6):
>
> $err_{S}\leq err_{T}+LA+FA$
>
> where the label alignment (LA) term is given by
>
> $LA=-\frac{1}{N}\sum_{i}\log\Big(\frac{p_s(y_i\mid z_i)}{p_T(y_i\mid z_i)^{\kappa(y_i,z_i)}}\Big)$
>
> Here, the transportation kernel $\kappa(y_i,z_i)$ arises from the decomposition of the generalization error and serves as a distributional correction term, rather than a sample-level reweighting function. It can be estimated using training data and the teacher’s predictive distribution (see Section D).
>
> That is, while confidence-weighted KD applies scalar weights outside the loss to emphasize certain samples, our method introduces a multiplicative correction inside the log-density ratio, fundamentally altering the distribution being matched. This reflects a shift from instance-level re-weighting to distribution-level transport alignment.
> In this sense, UCMKD is not a variant of confidence-weighted KD in paired settings, but a theoretically grounded framework for unpaired cross-modal distillation.
>
> [1] P. Chen et al. Distilling Knowledge via Knowledge Review. CVPR, 2021.
>
> [2] Liu et al. NORM: Knowledge Distillation via N-to-One Representation Matching. ICLR 2023.
>
> [3] Y. Zhang et al.Prime-Aware Adaptive Distillation. ECCV 2020
>
> [4] Su et al. EA-KD: Entropy-based Adaptive Knowledge Distillation. ICCV 2025
>
> [5] Iliopoulos et al. Weighted Distillation with Unlabeled Examples. NeurIPS 2022.

---

> > ### Author Rebuttal · Reviewer_zKfX · 2026-04-03
> >
> > Thank the reviewer for the detailed response. All of my concerns have been addressed, and I have therefore adjusted my score accordingly. By the way, the quality of this paper could be further improved by enhancing the visual presentation of the figures.

---

> > > ### Author Response · Authors · 2026-04-06
> > >
> > > We sincerely thank the Reviewer for acknowledging that the concerns are fully addressed and for raising the score. We also appreciate the Reviewer’s suggestion to enhance the visual presentation of the figures. We will update the figure in the revised version of this paper.

---

### Official Review · Reviewer_mgJo · 2026-03-13

**Soundness:** 4
**Presentation:** 3
**Significance:** 3
**Originality:** 3
**Overall Recommendation:** 5
**Confidence:** 4

**Summary:**

The subject of this paper is Cross-Modal Knowledge Distillation (CMKD), a process where a sophisticated "teacher" model trained in one modality, such as audio, transfers its intelligence to a "student" model in a different modality, such as video. The central issue identified by the authors is that existing CMKD methods rely heavily on paired or aligned multimodal data, where each data point in the teacher's modality must have a direct counterpart in the student's modality. Because synchronized datasets are often expensive, or even impossible to collect in real-world scenarios, the scalability of traditional distillation is severely limited. To solve this, the authors propose a framework called UCMKD that enables knowledge transfer using entirely unpaired data by aligning the underlying distributions of the modalities rather than individual samples.

The authors' method is inspired by a theoretical analysis that decomposes the gap between models into "feature alignment", which synchronizes latent representations, and "label alignment", which aligns predictive outputs. Their proposed algorithm implements this via a bi-level optimization process. In the first stage, the Wasserstein distance is used to minimize the discrepancy between representation distributions, while the second stage employs a specialized "label transport kernel" to selectively distill teacher knowledge that is most relevant to the student's task. The authors show experimental results of their approach on several datasets using ResNet networks.

**Compliance With Llm Reviewing Policy:**

Affirmed.

**Final Justification:**

Overall, I am raising my score because I do appreciate the detailed response of the authors (to me and other reviewers), but I feel the paper could benefit from stronger experimental evidence (mainly larger scale).

**Key Questions For Authors:**

I think it would be very helpful for the paper if the authors could show evidence that their approach is scalable in real-world scenarios with modern models much bigger than ResNets instead of leaving this as future work.

**Limitations:**

See weaknesses above.

**Strengths And Weaknesses:**

Strengths

— The authors identify an interesting problem, and give a sound analysis.

— Novel generalization bounds.

Weaknesses

— The main weakness of the method is that the bi-level optimization process used seems computationally expensive and seems to me that it might not be scalable in real-world scenarios. Indeed, the authors mainly demonstrate experiment with ResNet 50 and ResNet 18 networks, which in the age of LLMs seem particularly small.

---

> ### Author Rebuttal · Authors · 2026-03-31
>
> We sincerely thank the Reviewer for the thoughtful and detailed review, as well as the accept-leaning recommendation. We also appreciate the Reviewer’s constructive feedback, which are addressed below.
>
> >The main weakness of the method is that the bi-level optimization process used seems computationally expensive and seems to me that it might not be scalable in real-world scenarios.
>
> As we acknowledged in Section 6, the bilevel optimization introduces additional training cost. However, this overhead does not change the scaling behavior with respect to dataset size and remains comparable to lightweight baselines such as vanilla feature-based KD (i.e., Feature Distill), which provides a conservative reference point for processing time comparison.
> The additional cost amounts to only a constant-factor increase in forward/backward passes and does not introduce any additional dependence on the number of training samples, thus preserving KD’s scalability in real-world scenarios. We show this empirically and theoretically below.
> Empirically, we observe that the per-epoch training time increases by a small constant factor (typically between 1.2× and 2.9× across datasets, as shown below), which we consider acceptable given the significant performance gains over state-of-the-art baselines (see Table 1 and Table 2 in the main text).
>
> | Method|AVE||Cremad||Ravdess||Vggsound||
> |------------------------|--------------|--------------|--------------|--------------|--------------|--------------|--------------|--------------|
> |                        | A→V          | V→A          | A→V          | V→A          | A→V          | V→A          | A→V          | V→A          |
> | UCMKD       | 31.8s        | 44.13s       | 34.8s        | 63.6s        | 103.3s       | 83.8s        | 748.2s       | 692.3s       |
> | Feature distill | 14.7s        | 16.3s        | 22.5s        | 31.9s        | 59.4s        | 53s          | 624.8s       | 237.9s       |
> | Relative Overhead | 2.16x        | 2.76x        | 1.54x        | 1.99x        | 1.73x        | 1.58x        | 1.20x        | 2.91x        |
>
> Theoretically, we provide below the complexity analysis of UCMKD compared to feature-based KD. Let $N$ be the number of data points and consider the same backbone across methods. Let $F_1, F_2$ denote the forward cost of the student and the teacher model. Let $B_1$ denote the backward cost of the student model and let $W(N)$ denote the cost of computing the Wasserstein distance (via Sinkhorn w/ entropic regularization).
>
> For feature-based KD, the cost per iteration is:
>
> $C(FKD) =N*(F_1 + F_2 + B_1) + W(N) $
>
> For our method (UCMKD) with one feature-alignment step and one label-alignment step per iteration (as used in our experiments):
>
> $C(Our) = 3N*(F_1 + B_1) + N*F_2 + W(N) $
>
> As such, we have $C(Our) < 3C(FKD)$. The complexity analysis essentially shows that UCMKD introduces only a constant-factor overhead in the number of forward/backward passes while preserving the optimal transport cost W(N), which is often the main contributor governing scaling with respect to dataset size. With $\epsilon$-entropic regularization, the Wasserstein distance can be efficiently computed via the Sinkhorn algorithm[1].
>
> > I think it would be very helpful for the paper if the authors could show evidence that their approach is scalable in real-world scenarios with modern models much bigger than ResNets instead of leaving this as future work.
>
> We thank the Reviewer for this helpful suggestion. To further evaluate scalability in more realistic settings, we provide additional experimental results using a ViT-based architecture, which is representative of modern large-scale vision models.
> | Method                  | AVE                  |              | Ravdess                  |              |
> |------------------------|----------------------|--------------|--------------------------|--------------|
> |                        | **A→V**              | **V→A**      | **A→V**                  | **V→A** |
> | Teacher            | 75.87                | 70.15        | 90.41                    | 89.11        |
> | CE                | 51.19                | 53.73       | 65.63                   | 66.13       |
> | Feature Distill        | 50.96                | 56.22       | 69.83                   | 67.73       |
> | UCMKD        | **56.97**               | **58.21**       | **80.32**                   | **72.43**       |
>
> The reported results show that when changing to a backbone network much bigger than ResNets, UCMKD still expectedly achieves the best performance across different datasets. Combining with the complexity analysis above, it shows overall evidence that our approach is scalable in real-world scenarios.
>
> [1] Peyré et al. Computational optimal transport, 2020

---

> > ### Author Rebuttal · Reviewer_mgJo · 2026-04-03
> >
> > I appreciate the authors' careful rebuttal.
> >
> > (1) "As such, we have $C(Our) < 3C(FKD)$. The complexity analysis essentially shows that UCMKD introduces only a constant-factor overhead in the number of forward/backward passes while preserving the optimal transport cost W(N), which is often the main contributor governing scaling with respect to dataset size. With $\epsilon$-entropic regularization, the Wasserstein distance can be efficiently computed via the Sinkhorn algorithm[1]."
> >
> > While I am sympathetic to theoretical arguments, a "constant factor" of overhead is non-negligible in practice — for example people tend to be very happy with a "10x-improvement".
> >
> > (2) It would be helpful if you gave more details about the  VIT model used.
> >
> >
> > Overall, I am raising my score because I do appreciate the detailed response of the authors (to me and other reviewers), but I feel the paper could benefit from stronger experimental evidence (mainly larger scale).

---

> > > ### Author Response · Authors · 2026-04-06
> > >
> > > We sincerely thank the Reviewer for the thoughtful follow-up and for increasing the score to 5. We address the remaining points below.
> > > >While I am sympathetic to theoretical arguments, a "constant factor" of overhead is non-negligible in practice — for example people tend to be very happy with a "10x-improvement".
> > >
> > > We agree that constant factors can be important in practice. Our intention was not to suggest that the overhead is negligible, but to emphasize that UCMKD does not introduce additional scaling relative to the optimal transport (OT) component, which often dominates the overall cost. As such, while the overhead is not negligible, it remains manageable in practice. We will revise the wording to better reflect this nuance, and agree that improving computational efficiency is an important direction for future work.
> > > >It would be helpful if you gave more details about the VIT model used.
> > >
> > > Thank you for the suggestion. We will include full details of the ViT models in the main text. Specifically, we use ViT-B (patch16-224, ~86M parameters) as the teacher and ViT-S (patch16-224, ~22M parameters) as the student, following the existing practice in [1-3].
> > > >I feel the paper could benefit from stronger experimental evidence (mainly larger scale).
> > >
> > > We agree that larger-scale evaluation would further strengthen the empirical study. In response, we have conducted additional experiments with a larger teacher model (ViT-L/16, ~300M+ parameters) while keeping the student the same. The results (see below) are consistent with our previous findings and support the scalability of UCMKD to larger model regimes. We will include these results and corresponding discussion in the final version. This parameter regime already reflects a practically relevant large-model setting, and scaling to even larger (e.g., billion-parameter) models is a natural direction for future work.
> > > | Method     | AVE                  |              | Ravdess              |              |
> > > |------------|----------------------|--------------|----------------------|--------------|
> > > |            | **A→V**              | **V→A**      | **A→V**              | **V→A**      |
> > > | **Teacher**| 72.637               | 76.87        | 95.50                | 90.57        |
> > > | **CE**     | 51.19                | 53.73      | 65.63                | 66.13       |
> > > | **Feadistill** | 56.468           | 54.478       | 74.53                | 62.24        |
> > > | **UCMKD**| 59.453               | 61.194       | 88.42                | 77.42        |
> > >
> > > In addition, we observe that the runtime remains practical at this larger scale. In particular, UCMKD requires approximately 400s vs 149s per epoch for A→V, and 116s vs 45s for V→A, respectively, compared to Feature Distillation, reflecting an affordable constant-factor overhead (2.6x-2.7x) which is consistent with our earlier analysis.
> > >
> > > We thank the reviewer again for the constructive feedback.
> > >
> > > [1] Kai Wang et al. Attention  Distillation: self-supervised vision transformer students need more guidance.
> > >
> > > [2] Sravanti Addepalli et al. Leveraging Vision-Language Models for Improving Domain Generalization in Image Classification. CVPR 2024
> > >
> > > [3] Qian Zhang et al. CLIP-SDMG: CLIP knowledge distillation based on semantic decoupling and mask generation.
> > >
> > >
> > > [*] Note that CE is a no-distillation baseline and therefore it remains unchanged across different teacher models.

---

### Decision · Program_Chairs · 2026-04-30

**Decision:**

Accept (regular)

**Comment:**

This paper studies the cross-modal knowledge distillation without paired data. During rebuttal, the reviewers generally agreed the paper's theoretical framing and empirical scope. The authors also addressed most concerns during rebuttal with additional empirical results. There are some concerns that require further revision, e.g., the methodological novelty is moderate to some extent, and the main unpaired setting seems idealized as it is mainly simulated from paired datasets, and the method might lead to nontrivial computational cost. Overall, I recommend accept.